# Uncovering supramolecular chirality codes for the design of tunable biomaterials

Stephen J. Klawa [1], Michelle Lee [2], Kyle D. Riker [1], Tengyue Jian[1,3], Qunzhao Wang[1], Yuan Gao[1], Margaret L. Daly [1], Shreeya Bhonge[1], W. Seth Childers [2,4], Tolulope O. Omosun[2,5], Anil K. Mehta [2,6], David G. Lynn [2,7] ✉ & Ronit Freeman [1] ✉

In neurodegenerative diseases, polymorphism and supramolecular assembly of β-sheet amyloids are implicated in many different etiologies and may adopt either a left- or right-handed supramolecular chirality. Yet, the underlying principles of how sequence regulates supramolecular chirality remains unknown. Here, we characterize the sequence specificity of the central core of amyloid-β 42 and design derivatives which enable chirality inversion at biologically relevant temperatures. We further find that C-terminal modifications can tune the energy barrier of a left-to-right chiral inversion. Leveraging this design principle, we demonstrate how temperature-triggered chiral inversion of peptides hosting therapeutic payloads modulates the dosed release of an anticancer drug. These results suggest a generalizable approach for fine-tuning supramolecular chirality that can be applied in developing treatments to regulate amyloid morphology in neurodegeneration as well as in other disease states.

Helical chirality with defined left- or right-handed supramolecular twists is an abundant motif in living systems, guiding function and dysfunction[1-3]. These supramolecular helical structures are stabilized by an array of weak interactions such as hydrogen bonding, electrostatic interactions, π-π interactions, and van der Waals forces, and their transient nature allows for chiral conversion between right- and left-handed forms to achieve specific biofunctionality[3]. For example, gene expression is regulated through a transient inversion from a right-handed B-DNA to a left-handed Z-DNA which enables transcription factors to bind[4]. The bacterial flagella can undergo a chiral inversion of its helical architecture to alternate between running and tumbling modes[5]. In yeast, β-amyloids with pH-induced chirality inversion serve as heritable genetic elements or infectious prions which affect various pathways from regulating growth rates to triggering cell death[6-12]. Interestingly, HET-s prions formed at conditions which should favor right-handed twists (pH 7) are more infectious than

the left-handed prions formed at pH 2[12-14]. In humans, predominantly right-handed amyloid assemblies have been implicated in Alzheimer's disease (AD) and shown to exhibit a higher resistance to proteinase K degradation, versus their left-handed counterparts[15-18]. This property could be associated with disease progression and amyloid accumulation in plaques[18]. Despite extensive research into amyloid aggregation, it remains unclear what disease conditions guide left versus right helical chirality, what might facilitate a chiral inversion, and how the helical chirality is impacted by different variations in the primary sequence[15-20].

Studies of the AD-associated amyloid-β42 (Aβ42) core self-assembly motif (Aβ16-22, KLVFFAE) revealed that protonating the E amino acid directs left-handed supramolecular nanotubes by attenuating the inter-strand K16 and E22 interactions in anti-parallel β-sheet fibrils formed at neutral pH[21,22]. Replacing the glutamate with leucine resulted in pH-independent left-handed supramolecular nanotube

[1]Department of Applied Physical Sciences, University of North Carolina, Chapel Hill, NC 27599, USA. [2]Department of Chemistry, Emory University, Atlanta, GA 30322, USA. [3]Broad Pharm, San Diego, California 92121, USA. [4]Department of Chemistry, University of Pittsburgh, Pittsburgh, PA 15260, USA. [5]U.S. Department of Justice, Chicago, IL 60603, USA. [6]The National High Magnetic Field Laboratory, University of Florida, Gainesville, FL 32611, USA. [7]Department of Biology, Emory University, Atlanta, GA 30322, USA. ✉e-mail: dlynn2@emory.edu; ronifree@email.unc.edu

assemblies with narrower nanotubes (38 nm vs 52 nm diameter) and twists with only a 13° angle from the nanotube axis compared to the 23° twist found in the original sequence[22,23]. While this shows that the twist can be affected by small variations in the amyloid sequence, it is unclear what structural parameters govern left- or right-handed morphology, both on the individual β-sheet level as well as the overall leaflet packing, or if and how a transition between left- and right-handed amyloid states can occur. Uncovering the general principles controlling the supramolecular chirality of amyloids could open strategies for the design of therapeutic materials able to regulate the helical morphology of the amyloid plaques to render them more susceptible to degradation.

These biological helical assemblies sparked an interest in the design of artificial systems with controllable supramolecular helicity and chiral properties for applications in multiple fields[24]. Various amyloid-derived peptide sequences have been examined for their ability to guide helical chirality[25]. For example, mutating Fmoc-FF to Fmoc-FW directs a change from left- to right-handed helical assemblies[26]. However, design rules to favor left- vs. right-handed twists and how to switch between them remain unclear. External factors such as light, solvents, metal ions, pH, and temperature have been explored to trigger inversion of supramolecular chirality of assemblies formed by organic molecules as well as various β-sheet-forming peptides[13,19,27–38]. In one example, Fmoc-FWK was shown to adopt either left- or right-handed helices depending on pH-induced charges on the lysine residue[27]. Temperature-induced helical inversion of supramolecular assemblies suggests that transitions between the left- and right-handed states are regulated by an energy barrier where one handedness is a kinetically trapped nucleation state while the other is a more thermodynamically favorable state[34,35,39]. Computational studies investigating the relative stabilities of left- and right-handed β-sheet twists in peptide assemblies have revealed the right-handed twists to be thermodynamically favorable with an energy barrier to transition from left- to right-handed states[40–42]. It remains unknown, however, what molecular features of the peptide sequence are responsible for defining the magnitude of the energy barrier for chiral inversion.

Here, we systematically vary the sequence of Aβ16-22 at its C-termini, N-termini, and β-sheet domains to examine their role on the resulting morphology and chirality under different solvents and assembly pathway conditions. Following the assemblies using electron microscopy, electron diffraction, circular dichroism, and NMR, we uncovered that the supramolecular chirality is highly sensitive to sequence modifications at the C-termini, while the overall morphology is more sensitive to variations of the N-termini. Moreover, we found that the energy barrier for a left-to-right helical inversion in response to a thermal stimulus can be tuned by modifications to the C-termini. By encapsulating anticancer drugs such as doxorubicin in the aromatic interior of left-handed helical ribbons, we could regulate its dosed release for effective killing of tumor cells through the thermally triggered helical inversion of the structures. The ability to tune the chirality of amyloid-derived peptides through modification to the C-termini may enhance our understanding of protein amyloidosis and provide key insights to treat neurodegenerative diseases through manipulation of their chirality. This strategy would also inform the design of synthetic materials and devices operating through chiral inversion.

## Results

### β-sheet nucleation tunes amyloid supramolecular chirality

We sought to exploit the remarkable structural diversity accessible via 2-step nucleation of peptide phases (i.e., the formation of a dynamic meta-stable phase prior to mature assembly) to access dynamic supramolecular functional materials. Self-assembly of the nucleating motif of Alzheimer's Aβ42 peptide (KLVFFAE, Fig. 1a, Supplementary Figs. 1-3) has three propagating growth planes: backbone H-bonding to

generate β-sheets, sheet stacking into cross-β fibers/leaflets and leaflet association for higher order assemblies, each having the potential to regulate assembled morphology[22,43–46]. For example, assembling under conditions where the terminal E residue is protonated or substituted with L yields homogeneous peptide bilayer nanotubes (40% acetonitrile, pH 2)[22]. Simply substituting the terminal E/L residue with V (KLVFFAV, Fig. 1a), which is the amino acid most commonly found in a β-sheet based on statistical predictions of protein secondary structure, forms nanotubes (Fig. 1b) with a strikingly different cross-sectional area[47]. The flattened width of the KLVFFAV tubes dried on EM grids was more than 5 times larger, 397 ± 52 nm relative to KLVFFAE at 72 ± 5 nm (Fig. 1b). Oriented electron diffraction (Fig. 1c) confirmed the same cross-β architecture and revealed an increased twist angle for these KLVFFAV cross-β nanotubes relative to KLVFFAE, from 22° to 34°.

FT-IR analyses (Fig. 1d) were similar for both the KLVFFAE and KLVFFAV nanotubes showing intense amide-I bands at 1627 cm$^{-1}$ and a weaker mode at 1693 cm$^{-1}$, characteristic of anti-parallel β-sheets[48]. To evaluate the strand registry, isotopic enrichments of [1-$^{13}$C]V18 and [$^{15}$N]A21 were incorporated into KLVFFAE, KLVFFAL, and KLVFFAV peptides and each was allowed to assemble under acidic conditions. Solid-state NMR $^{13}$C-$^{15}$N distance measurements using $^{13}$C{$^{15}$N}REDOR pulse sequences[49,50] defined distances confirming out-of-register antiparallel β-sheets (Supplementary Fig. 4) independent of nanotube size as defined by small angle X-ray scattering (SAXS) (Supplementary Fig. 5)[22]. Previous simulations characterizing possible registrations for the KLVFFAE peptide show that only one out-of-register conformation is likely, suggesting that since the core β-sheet domain remains unchanged, these assemblies would also have identical strand registry[51]. To further investigate how packing may be different between these peptides, we performed X-ray diffraction (XRD; Supplementary Fig. 6) which revealed very similar peaks of d-spacing for all three peptides. The sharp reflection at 4.8 Å arises from the peptides H-bonded within the β-sheet, while the peak around 10 Å corresponds to lamination of β-sheets; the KLVFFAV peptide showed a much broader peak around 10 Å which is likely due to loss of rigidity given the larger diameter of those nanotubes. Interestingly, the only spectroscopic signature differentiating the E/L tubes and the V tubes was circular dichroism (Fig. 1e and Supplementary Fig. 7). The characteristic β-sheet red-shifted transition at ~220 nm is negative for KLVFFAE and KLVFFAL but positive for KLVFFAV, corresponding to left-handed and right-handed β-sheet axial twist respectively (for a detailed discussion defining β-sheet and supramolecular chirality within the context of this paper refer to Supplementary Note 1)[52]. As the handedness of the β-sheet is changed between these peptides but the registry of both the left-handed and right-handed tubes are the same, we suspect that β-sheet chirality is an independent parameter likely defined during the initial condensate nucleating step of the two-step nucleation process with the only difference in molecular conformation arising from the changes in Φ and ψ torsion angles needed to form a left-handed β-sheet or a right-handed β-sheet[40].

To evaluate this differential nucleation of left- and right-handed nanotubes, we explored the ability of KLVFFAL and KLVFFAV to co-assemble. The two peptides were mixed at a 1:1 molar ratio and their assembly was followed over time. After 1 month, a heterogeneous mixture of filaments and sheets was apparent by transmission electron microscopy (TEM; Supplementary Fig. 8). After further incubation, populations of narrow tubes and wider tubes along with a smaller population of short filaments were apparent by TEM (Supplementary Fig. 8). CD measurements of the aged co-assembly appear as a 1:1 summation of the KLVFFAL and KLVFFAV CD curves (Supplementary Fig. 9).

Further quantitative results were obtained by spin counting $^{15}$N dephasing of proximal $^{13}$C spins with $^{13}$C{$^{15}$N}REDOR solid-state NMR. This took advantage of the V18 backbone H-bond with A21 on the adjacent β-strand and involved a different isotope enrichment of [1-$^{13}$C]

A21 and [$^{15}$N]V18 on KLVFFAL and the methyl of [1-$^{13}$C]A21 enriched in KLVFFAV (Supplementary Fig. 9). In the co-assembly, 70% of A21 carbonyls are dephased by (hence in proximity to) $^{15}$N in the $^{13}$C{$^{15}$N} REDOR experiments and correspondingly, 30% of the A21-methyls are dephased (Supplementary Fig. 10). The 30% A21-methyl dephasing could result for a range of co-assemblies (Supplementary Fig. 10). Given the far weaker ellipticity of amyloid fibers relative to nanotubes, we assign the observed ellipticity as derived dominantly by almost equal populations of the large and small nanotubes. Therefore, there are at least three distinct nucleation events, pure L and V nanotubes and the fibril co-assemblies which account for the mixed fibers. The fact that the difference of a single CH$_2$ in the C-terminal amino acid can so strongly select for nucleation and propagation of supramolecular chirality is striking, but this single observation does not explain why this occurs to enable peptide design.

We expect such distinct nucleation events to be both solvent and temperature dependent. To further investigate the interactions responsible for this nucleation, we generated the peptide KLVFFA-PEG$_2$ which removes the electrostatic interactions from KLVFFAE but remains hydrophilic and lacks the additional β-sheet interactions of KLVFFAV (Fig. 2a and Supplementary Fig. 11). Both KLVFFAE and KLVFFAV assembled into ribbon-like structures after 2 hours in 10% hexafluoroisopropanol (HFIP), while KLVFFA-PEG$_2$ showed only tiny aggregates (Fig. 2b). We presume that the growth of the KLVFFAE peptide into long fibrils is likely a result of electrostatic interactions as the HFIP is not pH adjusted. We next annealed these peptides (heated to 95 °C and cooled at 1 °C/min) which resulted in bundles for the

KLVFFAE and nanotubes for KLVFFAV; KLVFFA-PEG$_2$ did not assemble into amyloid-like structures during this process, highlighting the role of C-terminal amino acid interactions in the assembly process (Fig. 2c).

Interestingly, the CD spectra of the aged assemblies (Fig. 2d and Supplementary Fig. 12) revealed that both KLVFFAE and KLVFFAV were left-handed β-sheets in these conditions, suggesting that we had induced the KLVFFAV peptide into a trapped state as compared to the ACN prep. However, after annealing, only the KLVFFAV CD spectra inverted to show a right-handed β-sheet (Fig. 2e). We presume that the combination of reduced hydrophobic interactions and stronger β-sheet interactions in the ACN prep with higher organic solvent content is responsible for allowing the formation of the tubes at room temperature in 40% ACN, while in 10% HFIP, heating is required to overcome the energy barrier of the kinetically-trapped product and enable stable right-handed nucleation[51]. We further investigated the process of the chirality inversion by monitoring CD during the annealing process (Supplementary Fig. 13); at 85 °C, the β-sheet signal remained left-handed but transitioned to right-handed during the slow cooling process. The slow cooling process was shown to be vital for the chirality inversion and formation of tubes, as a solution that was cooled to room temperature in an unregulated manner remained trapped in the left-handed state and did not form tubes as there was insufficient time to form a right-handed nucleation point (Supplementary Fig. 14). However, we showed that if the β-sheets are already nucleated in the right-handed state, unregulated rapid cooling will allow nanotubes to reform after melting (KLVFFAV in 40% ACN), but if they are left-handed (KLVFFAE, in 40% ACN), they will not reform (Supplementary Figs. 15,

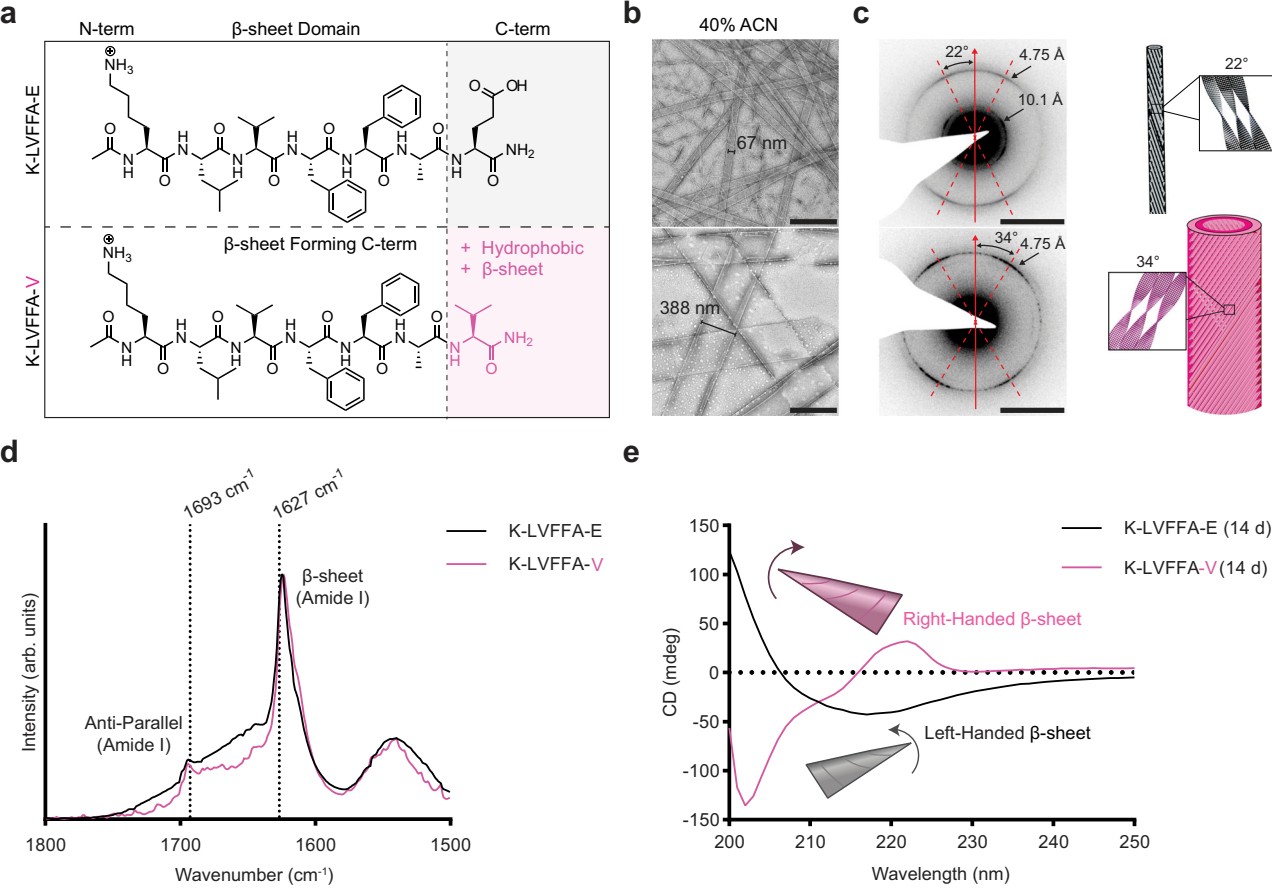

**Fig. 1 | C-terminal amino acid directs handedness of β-sheet twist.** Chemical structures (**a**) showing the replacement of E with V on the C-termini. TEM images (**b**) of peptides assembled in 40% ACN, 0.1% TFA for 2 weeks (representative of 3 independent assemblies, scale bars = 500 nm). Electron diffraction (**c**) (scale bars = 2 nm$^{-1}$) and schematics showing the twist angle of the tube (representative image of diffraction patterns taken on 3 different areas of the grid). FT-IR (**d**) of KLVFFAE and KLVFFAV peptides assembled in 40% ACN, 0.1% TFA for 2 weeks showing anti-parallel β-sheet signal. CD spectra (**e**) of KLVFFAE and KLVFFAV peptides assembled in 40% ACN, 0.1% TFA for 2 weeks. In schematics and graphs, KLVFFA-E = black/gray, KLVFFA-V = pink.

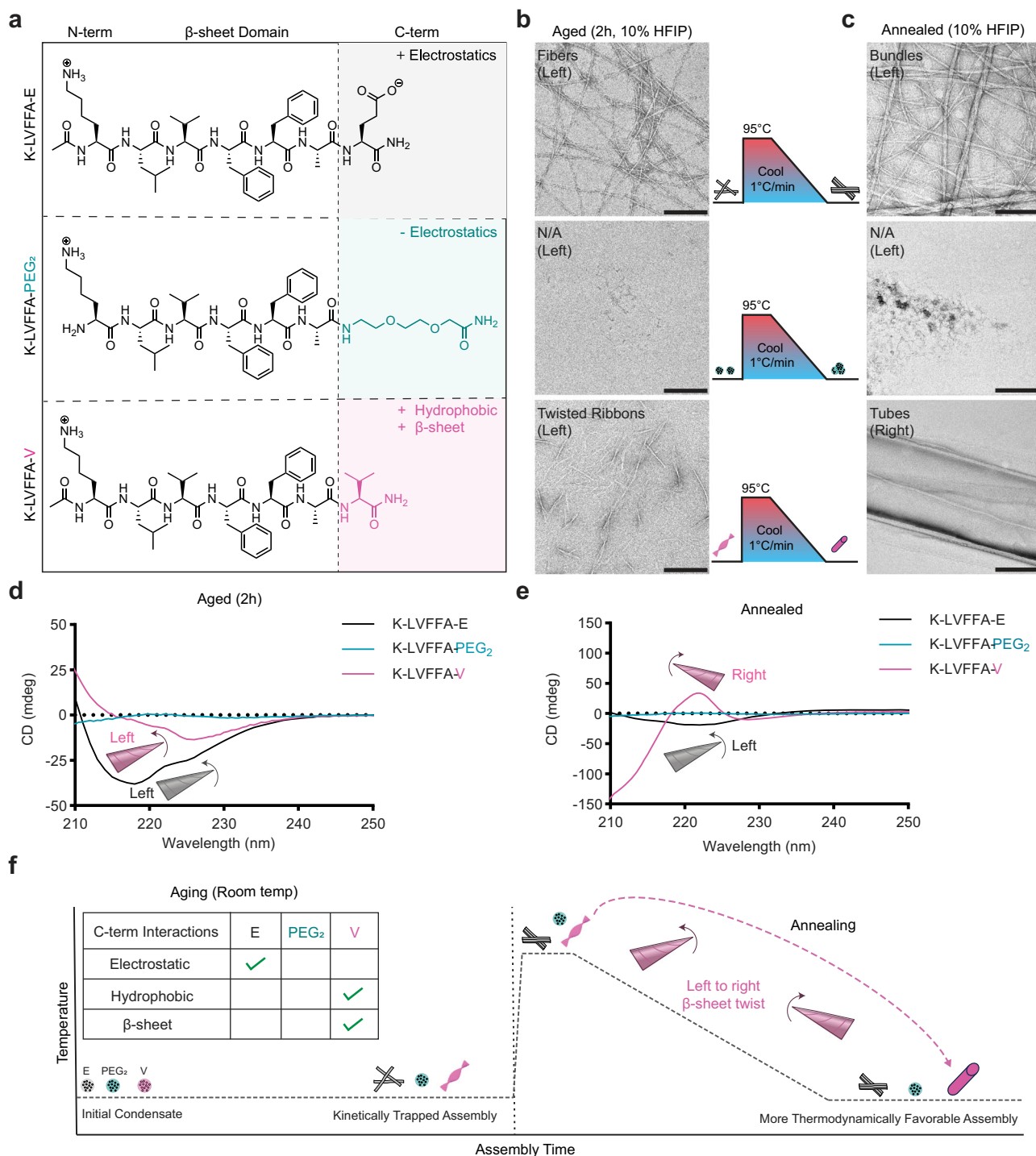

**Fig. 2 | Tuning driving forces for assembly nucleation directs accessible chirality.** Chemical structures (**a**) showing the mutations to C-terminal interactions. TEM images (**b**) of peptides assembled in 10% HFIP for 2 h (scale bars = 200 nm). TEM images (**c**) of peptides annealed in 10% HFIP for 2 h (scale bars = 200 nm). All TEM images are representative of at least two separate assemblies. CD Spectra (**d**) showing the peptides aged for 2 h in 10% HFIP. CD spectra (**e**) showing peptides after annealing in 10% HFIP. Schematic (**f**) of assembly pathway and driving forces responsible for morphologies at both the kinetically trapped states and more thermodynamically favorable states. In schematics and graphs, KLVFFA-E = black/gray, KLVFFA-PEG₂ = teal, KLVFFA-V = pink.

16). As predicted, both solvent and temperature can be used to preferentially direct the nucleation and growth of right- and left-handed β-sheet assemblies.

## Regulation of supramolecular morphology

The increased β-sheet propensity of valine and the evidence for facial complementarity in cross-β assembly suggested that sidechain packing interactions could significantly impact left- vs. right-handed β-sheet nucleation[22]. Phenylalanine not only increases the aromatic interactions, but also experimental observations measuring the ΔΔG of β-sheet formation with various amino acids suggests that from a thermodynamic standpoint phenylalanine may be the amino acid with greatest propensity for cross-β assembly[53,54]. As an initial test of tuning the assembly, the FF dyad of the KLVFFA-PEG₂ peptide was extended to

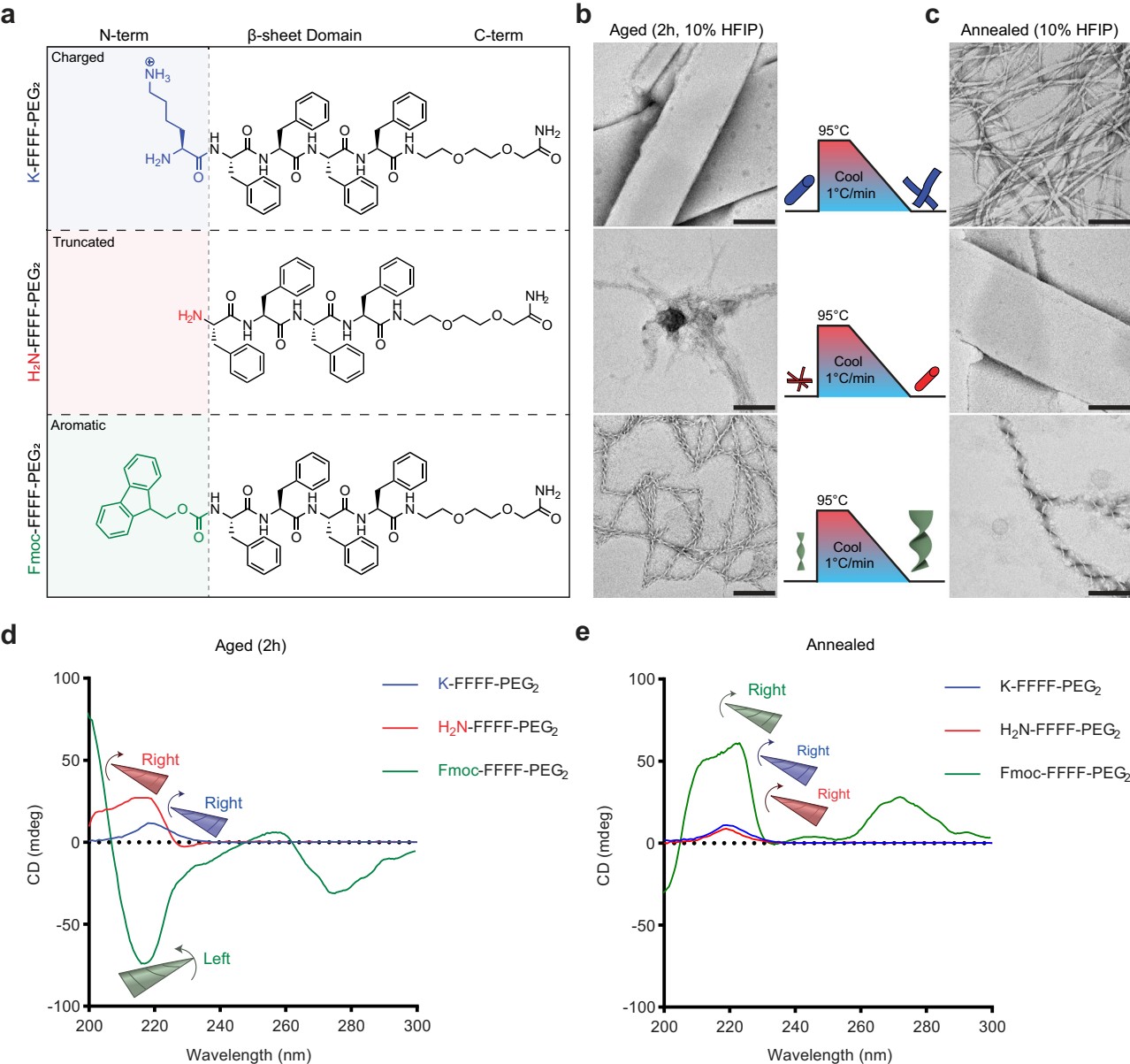

**Fig. 3 | N-terminal Fmoc modification induces kinetic trap enabling chirality inversion.** Chemical Structures of peptides (**a**) with mutations to the N-terminus. TEM images of peptides aged 2 h (**b**) or heated to 95 °C and cooled to room temperature (**c**) in 10% HFIP (all scale bars are 200 nm, images are representative of multiple images from a single assembly except for Fmoc-FFFF-PEG₂ which was repeated more than 3 times with all assemblies being reproducible). CD spectra (**d**) of the aged peptides shows that the Fmoc induces left-handed β-sheets. CD spectra of the annealed peptides (**e**) shows that all are right-handed indicating a chirality inversion of Fmoc-FFFF-PEG₂. For schematics and graphs, K-FFFF-PEG₂ = blue, H₂N-FFFF-PEG₂ = red, Fmoc-FFFF-PEG₂ = green.

four phenylalanine residues to give K-FFFF-PEG₂. For this tetraphenylalanine domain, we aimed to explore the effect of the N-terminal interactions (Fig. 3a, Supplementary Figs. 17-19) on this assembly by either removing the terminal lysine residue or replacing it with the aromatic capping group, Fmoc, a common hydrophobic modification that also removes the N-terminal charge[55,56]. TEM images after 2 hours of assembly (Fig. 3b) gave distinct morphologies for all three modifications: K-FFFF-PEG₂ formed nanotubes, H₂N-FFFF-PEG₂ formed bundled fibrils, and Fmoc-FFFF-PEG₂ formed twisted ribbons with M-type (left-handed) supramolecular helicity. More remarkable, a heating/cooling cycle gave radically different morphologies for each peptide (Fig. 3c): K-FFFF-PEG₂ gave ribbons, H₂N-FFFF-PEG₂ formed tubes similar to the room temperature assembly of K-FFFF-PEG₂, and Fmoc-FFFF-PEG₂ formed larger ribbons which exhibit P-type helicity as observed on TEM.

The enhanced interactions of the β-sheet domain in K-FFFF-PEG₂ which resulted in tubes after 2 h in 10% HFIP is a stark contrast to the minimal assembly that was observed before the phenylalanine substitution. CD spectra (Fig. 3d, e) revealed that the β-sheet signal was right-handed for both the aged 2 h and annealed cases of K-FFFF-PEG₂ showing that these phenylalanine residues enabled the β-sheets to form a more thermodynamically favorable conformation. When the lysine is removed, the aggregated fibrils are likely due to the increased hydrophobicity of the molecule, but the fibrils are able to transform into tubes after an annealing cycle; again, the CD spectra revealed right-handed β-sheets for both the aged and annealed cases. However, when the Fmoc group is appended to the N-terminus, the M-type helices observed by TEM displayed a left-handed β-sheet CD signal at room temperature but after annealing the CD spectra inverted to be right-handed β-sheet, correlating with the

supramolecular chirality inversion from an M-type helix to a P-type helix.

To test if the supramolecular chirality inversion was induced through chiral amplification of the β-sheet to the supramolecular ribbons, we synthesized a variant of Fmoc-FFFF-PEG₂ where we replaced the L-amino acids with D-amino acids (Supplementary Fig. 20) which should reverse the chirality on the β-sheet level. As expected, assembling under the same conditions yielded right-handed β-sheets before annealing and left-handed β-sheets after annealing as seen on CD (Supplementary Fig. 21). As observed with TEM and SEM (Supplementary Fig. 21), this correlated with small P-type twisted ribbons before annealing and large M-type twisted ribbons after annealing. These ribbons were morphologically similar to the structures observed with the L-amino acid peptide except with reversed chirality on all levels showing that the supramolecular chirality obtained is a result of chiral amplification from the β-sheet.

To better understand the positional effects of the side chains in the case of Fmoc-FFFF-PEG₂ which enabled an inversion of both β-sheet twist and supramolecular chirality, we shortened the phenylalanine domain. Fmoc-FFF-PEG₂ (Supplementary Fig. 22) assembled with left-handed β-sheets both before and after the annealing cycle (Supplementary Fig. 23). In contrast, shortening to Fmoc-FF-PEG₂ (Supplementary Fig. 24) gave a right-handed assembly (P-type twisted ribbon) for both the aged and annealed states (Supplementary Fig. 25), like that observed with four phenylalanine residues after annealing. Similar to how side chain packing complementarity can dictate morphology, this shows that registry complementarity can also be used to influence supramolecular chirality.

## C-terminal modifications can tune dynamic supramolecular chirality inversion

The observation that the Fmoc-FFFF-PEG₂ peptide can form both left- and right-handed β-sheets correlating with M-type and P-type chiral nanostructures focused our attention on complementary C-terminal modifications to tune the nucleation. Replacement of PEG₂ with a terminal amide (Fmoc-FFFF-NH₂, Supplementary Fig. 26) gave fibrils with left-handed β-sheet CD signal after 2 hours (Supplementary Fig. 27). Annealing with the same heating/cooling cycle gave large ribbons but no chirality inversion (Supplementary Fig. 27). Previous reports suggested that increasing hydrophobicity of C-terminal amino acids in amyloid peptides inhibits supramolecular twist, so we recognized the importance of the PEG₂ motif and investigated further additions rather than replacement of this motif[57]. Either an azide-lysine (PEG₂-Kaz) or a cysteine (PEG₂-Cys) was added to enable further functionalization (Fig. 4a and Supplementary Figs. 28–29). As with PEG₂ alone, these peptides exhibited M-type supramolecular chirality by SEM (Supplementary Fig. 30) and TEM (Fig. 4b) and left-handed β-sheet chirality on CD (Fig. 4c) but inverted to right-handed β-sheet chirality and P-type twisted ribbons after the annealing cycle.

Remarkably, the inversion temperature of each peptide was distinct (Supplementary Fig. 31) and when plotted using the π-π* ellipticity transition at 273 nm (Fig. 4d), these inversion temperatures vary by more than 10 °C from the PEG₂ only peptide and fall within a biologically relevant range of 22–45 °C. The heterogeneity of the PEG₂-Kaz peptide on TEM is likely impacted by an inversion temperature (22.7 °C) which is very near that of room temperature. To determine if the higher inversion temperature of PEG₂-Cys (45.1 °C) is due to oxidation to form disulfide bonds, we measured the inversion temperature in the presence of TCEP (Supplementary Fig. 32) which showed very little change to 47.3 °C.

While the peptide variations are morphologically similar, we observed that the ribbon width due to lamination increased slightly from the azide-modified peptide to the PEG₂ or Cys peptides (Fig. 4e). We attempted to correlate these changes in lamination to nucleation rates which may favor different growth planes. To investigate how

initial nucleation varied between these peptides, we tracked CD measurements of the β-sheet signal over time. When stock peptide solutions are dissolved in HFIP, a very weak CD signal is observed, but almost immediately after the HFIP solution is added to water, strong β-sheet characteristics are obtained. These β-sheet signals then continue to grow until saturation (Fig. 4f). Fitting this data to a two-phase exponential decay model revealed that the nucleation rate also increased from the azido peptide to the cysteine peptide and correlated well with transition temperature (Fig. 4g). This data suggests that the initial lamination growth plane may be a kinetically favorable state which peptides favor when nucleating rapidly while peptides that nucleate slower are able to favor other growth planes.

Each of a peptide's 2-step nucleation events is environmentally sensitive, and the intermediate condensate can be self-selected as well as externally templated into different polymorphs[58–61]. Two limiting models might be considered for these conformational inversions. The first involves a nucleus stabilized in the initial biomolecular condensate that is sufficiently destabilized when propagating outside of the condensate to allow the paracrystalline nucleus to access a different supramolecular structure during propagation. An alternative might involve the M-type structures cooperatively untwisting and retwisting with the opposite twist. We first attempted to track this transition using TEM. To do this, we assembled the Fmoc-FFFF-PEG₂ peptide and equilibrated it to various temperatures above and below the transition temperature before preparing TEM grids. From TEM images at these temperatures (Fig. 5a), we observed that during this rearrangement the M-type helices appear to untwist with a slight increase in pitch (Supplementary Fig. 33). The fibers then appeared to align with each other before merging and re-twisting in the opposite direction to form larger P-type twisted ribbons.

The thickness of the ribbons in both the left- and right-handed states does not change significantly (Supplementary Fig. 34) and in both states is approximately the thickness expected from a bilayer of peptides which was additionally verified by electron tomography of the right-handed structures (Supplementary Fig. 35 and Supplementary Movie 1). Both states also exhibited anti-parallel β-sheet characteristics as determined by FTIR (Supplementary Fig. 36), with the annealed structures showing a higher frequency peak compared to the aged sample (1698 vs. 1687 cm⁻¹), likely arising from tighter hydrophobic packing driving more side-by-side lamination to form the larger right-handed assemblies[62].

While these results suggest that the fibers untwist and align during the chirality inversion, we wanted to investigate the possibility of right-handed nucleation points forming and propagating assembly. To test if the P-type helices could serve as nucleation points to direct a right-handed twist without additional heating, we annealed Fmoc-FFFF-PEG₂ and after the solution was cooled, the structures were sonicated to break them into small seeds. Monomers were added to these smaller seeds in varying ratios (Fig. 5b), and interestingly we found that by the addition of only 10% seeds, the resulting CD spectra is primarily right-handed (Fig. 5c). However, to completely regain the signal of only right-handed structures, 90% seeds needed to be used with only 10% free monomers. TEM images and quantification (Fig. 5d, e) align with the CD results showing some left-handed assemblies present until 90% seeds are used. To understand if this limitation was a result of maximum length of right-handed structures, we measured the length of the right-handed assemblies (Supplementary Fig. 37). This showed that while some growth occurs, the length cannot propagate to the original size before monomers become kinetically trapped in a left-handed state at which point they cannot be drawn to the right-handed seeds.

Together, these results suggest that multiple processes are occurring during chirality inversion; the inversion was then tracked in real time using confocal microscopy. We stained the structures with Thioflavin-T (ThT) and imaged structures as we heated the sample (Supplementary Movies 2,3). From the movie of a network

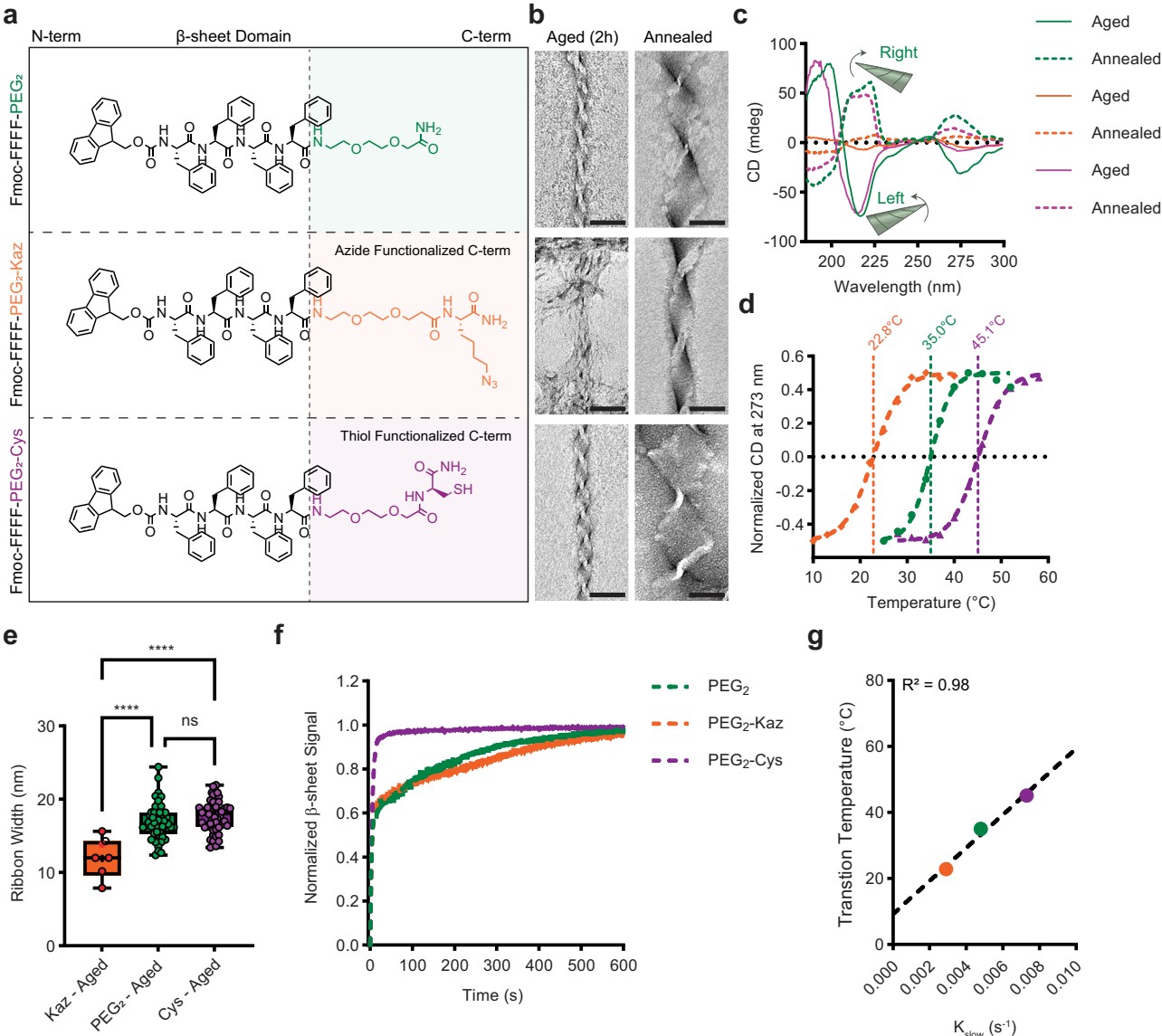

**Fig. 4 | C-terminal modifications enable a tunable energy barrier for chirality inversion.** Chemical structures (**a**) of peptides with added functional groups on the C-termini. TEM images (**b**) of all peptides before and after annealing showing supramolecular chirality inversion (scale bars = 50 nm, representative images from two independent assemblies). CD spectra (**c**) show that all peptides transition from left- to right-handed b-sheet twist after heating. Following CD spectra with temperature (**d**) revealed tunable transition temperatures of chirality inversion. Ribbon width (**e**) measured from TEM images for aged peptide assemblies (n = 6, 50, 50 for Kaz, PEG$_2$, and Cys respectively, differences were compared with a one-way ANOVA with Tukey's multiple comparisons test). Center line is median, the box extends to the 25th and 75th percentiles, and whiskers extend to the minimum and maximum data points (**** represents p values of 0.0000095 and 0.0000002 for PEG$_2$ and Cys respectively, ns represents a p value or 0.1307). Nucleation kinetics (**f**) measured by following CD signal at 216 nm. Correlation between nucleation rate and transition temperature (**g**) fitted with a linear regression. Peptides are color coded (orange diamonds = Kaz, green circles = PEG$_2$, and purple triangles = Cys).

(Supplementary Movie 2), we observed contraction of the network rather than disassembly and reassembly of structures, supporting our hypothesis that the structures can invert chirality without the need for disassembly. Furthermore, analysis of the width of the structures in Supplementary Movie 3 (Supplementary Fig. 38) revealed wider structures as temperature increased and structures laminate to form the right-handed structures. As the chirality inversion was observed with confocal microscopy, the dynamics of the material during heating using fluorescence recovery after photobleaching (FRAP) was explored. FRAP at temperatures below, at, and above the transition temperature (Supplementary Fig. 39) were fit to a one-phase association and the half-time of recovery at each temperature (Supplementary Fig. 40) showed that just before the transition temperature the half-time decreases as monomers become more mobile before increasing

dramatically at high temperatures, further indicating the stability of the right-handed β-sheet structures.

## Chirality inversion enables protease degradation
With the understanding of how the structure transition was taking place at these biologically relevant temperatures, we explored the stability of these supramolecular structures against degradation to assess their suitability for various biological applications. We chose to test stability against chymotrypsin as this protease is known to cleave peptide bonds after phenylalanine residues[63]. We exposed both the aged (M-type) and annealed (P-type) structures to chymotrypsin for 48 hours (Fig. 6a) and measured the degradation using a fluorescamine assay to quantify the amines produced by degradation (Fig. 6b and Supplementary Fig. 41). Interestingly, we observed that the P-type

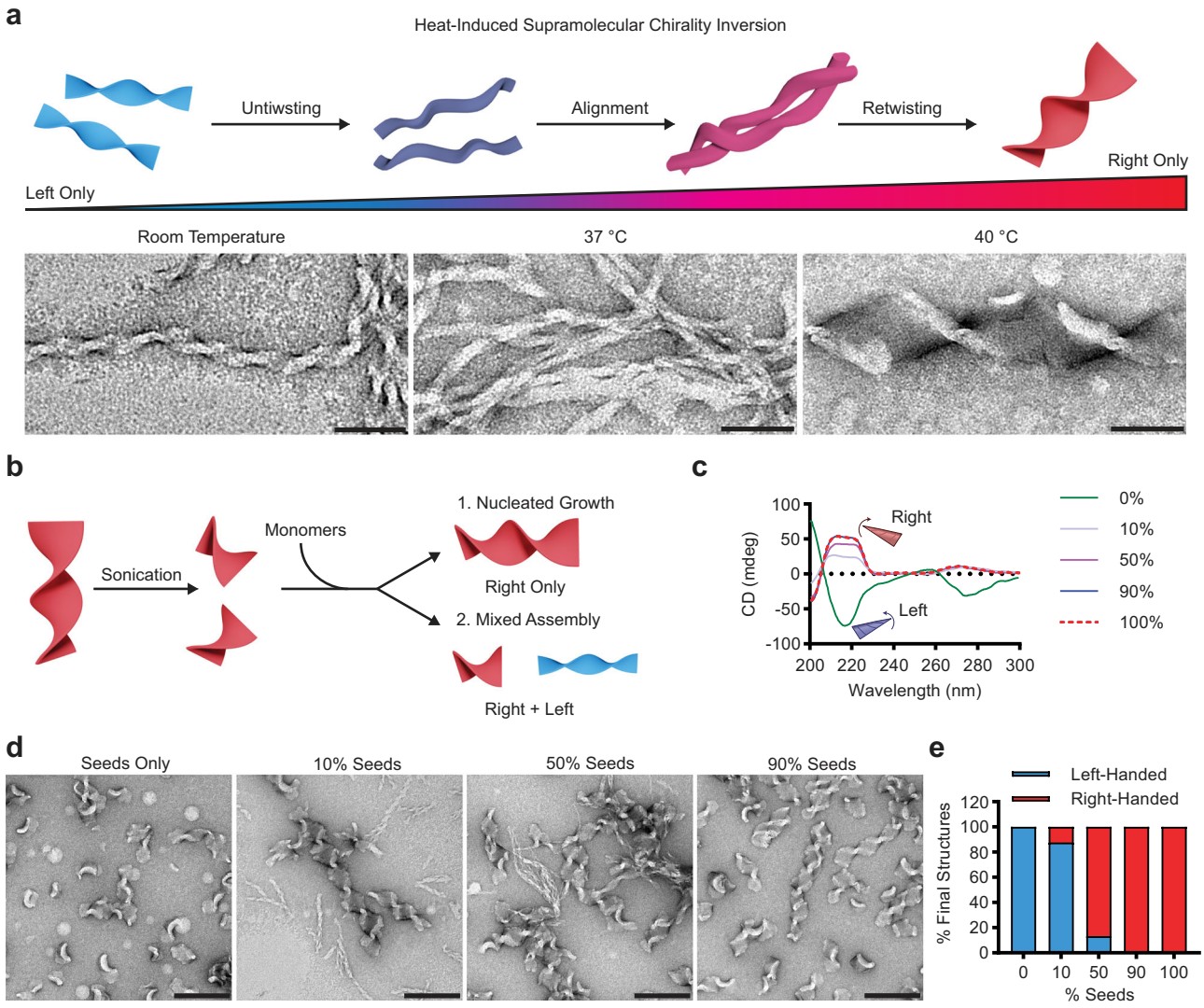

**Fig. 5 | Chirality inversion mechanisms.** Scheme and TEM images (**a**) of Fmoc-FFFF-PEG$_2$ prepared at RT, or heated to 37 and 40 °C to track the chirality inversion (50 nm scale bars). Schematic (**b**) of seeded growth via right-handed nucleation points. CD spectra (**c**) of nucleated assemblies grown with varying monomer:seed ratios. TEM images (**d**) of nucleated assemblies (200 nm scale bars). Quantification of number of left- and right-handed structures (**e**) ($n \geq 100$ structures for each case).

helices were more degraded than the M-type structures. To identify degradation products, analytical HPLC and MS were used to identify fragments after the assay (Fig. 6c and Supplementary Figs. 42–45). This analysis revealed that the M-type helices primarily were cleaved only once to remove the Fmoc-group, whereas the P-type helices were susceptible to cleavage of Fmoc-F from the N-terminal as well as further degradation with multiple cleavage sites. As the degree of degradation and byproducts were different for the two structures, we assessed whether the products had any impact on the activity of chymotrypsin. Using a colorimetric substrate, N-succinyl-L-phenylalanine-p-nitroanilide, we measured the remaining activity of chymotrypsin after 24 h of degrading either aged or annealed structures, showing that there was no significant impact on activity after degrading either structure (Supplementary Fig. 46). TEM images of the assemblies before and after degradation supported that the M-type ribbons (left-handed) remained mostly intact while the P-type ribbons (right-handed) appear much less uniform (Fig. 6d). These results suggest that the right-handed β-sheet packing, or torsion angles make the internal bonds more accessible to protease degradation compared to the left-handed state. The differences in protease stability between the two states provides potential for triggered degradation of the material as needed.

## A tunable drug delivery vehicle for efficient drug release

The ability of the structures to be degraded after chirality inversion suggests their benefits for drug delivery if inversion could be coupled to drug release and the vehicle could then be degraded and cleared after use. While we engineered our morphology-switching structures based on amyloid peptides derived from Alzheimer's disease, we attempted to utilize our materials as a delivery vehicle to target cancer cells. We suspected that the aromaticity of our peptides would be particularly useful for incorporating the anti-cancer drug doxorubicin (DOX) which has previously been mixed into self-assembled structures[64–66]. We hypothesized that if DOX was incorporated into the assembly that it may be released during the chirality inversion (Fig. 7a). To test the loading capacity of our peptides, we assembled our peptides in the presence of various DOX concentrations. Quantification of DOX remaining in solution after peptide assembly (Fig. 7b) revealed that when assembling with 250 μM peptide, maximal drug loading was achieved when peptides were assembled in the presence of 50 μM DOX, resulting in payloads of 68.8, 32.4, and 11.6 nmol DOX per μmol peptide for the PEG$_2$, PEG$_2$-Cys, and PEG$_2$-Kaz derivatives, respectively. Testing the incorporation of DOX within the same peptide but lacking the Fmoc group (H$_2$N-FFFF-PEG$_2$) established that the N-terminal Fmoc group is essential for DOX loading (Supplementary

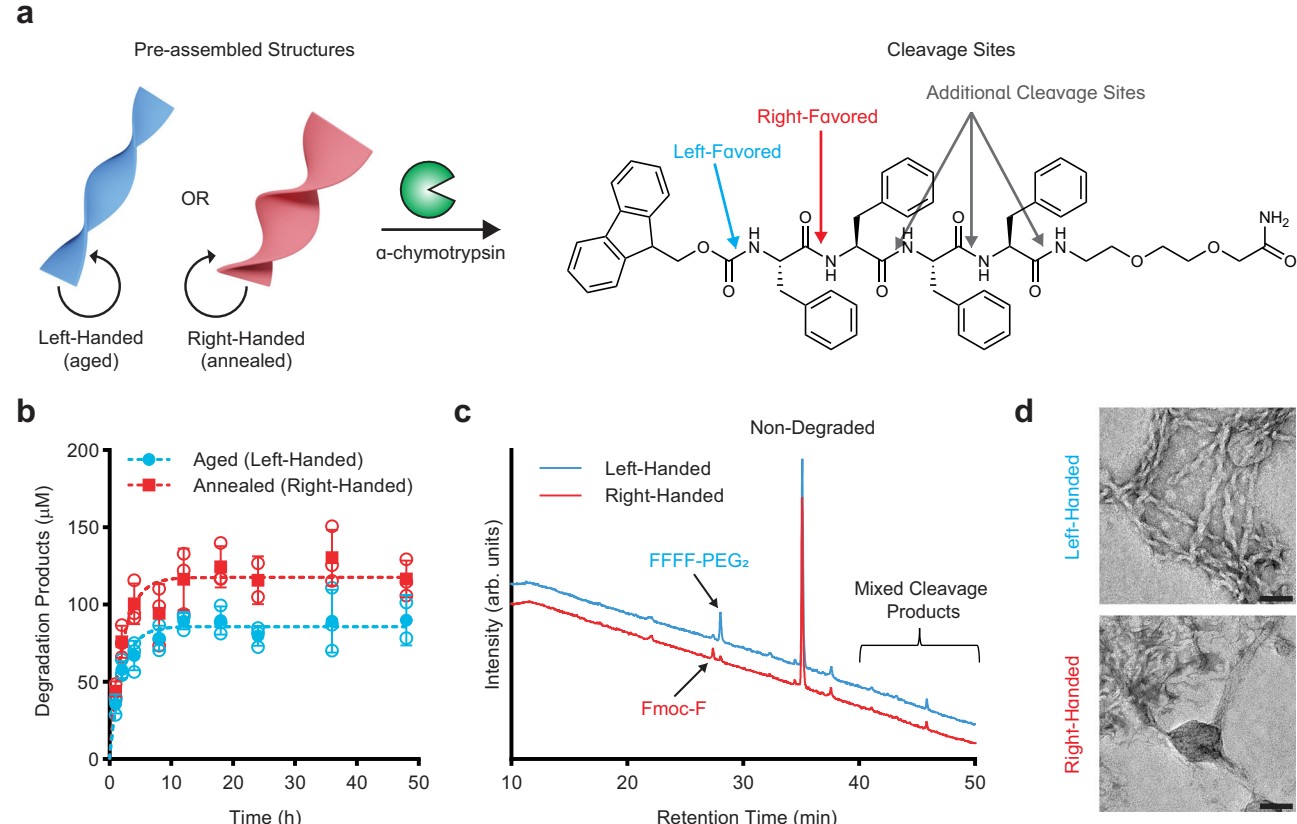

**Fig. 6 | Chymotrypsin digestion of structures.** Schematic of degradation assay (**a**) comparing left- and right-handed assemblies and favored cleavage sites identified on the chemical structure. Time course of degradation by chymotrypsin (**b**) quantified from a fluorescamine assay containing 50 μM peptide (250 μM potential degradation products) and 5 μg/mL chymotrypsin (*n* = 3 samples for each time point, open symbols indicate individual data points, solid symbols indicate mean with standard deviation error bars. Outliers removed with *Q* = 1%). HPLC traces (**c**) of peptides after 24 h degradation. TEM images (**d**) of aged and annealed samples after 24 h of chymotrypsin digestion (scale bars = 50 nm, representative images of a single grid for each sample). Blue represents left-handed structures; red represents right-handed structures.

Fig. 47). Since the loading efficiency decreases with the bulkiness of the C-terminal modification, the DOX likely forms aromatic interactions with the Fmoc group, but specifically packs at the bilayer interface of the anti-parallel strands where bulky C-termini modifications likely interfere with DOX incorporation.

To test the release of DOX, we took advantage of its native fluorescent properties. Using confocal microscopy, DOX fluorescence could be viewed within each peptide assembly. After heating to 37 °C for 1 h, a significant decrease in DOX intensity was observed for the $PEG_2$-Kaz and $PEG_2$ peptides, but the $PEG_2$-Cys exhibited no significant decrease in intensity (Fig. 7c, Supplementary Fig. 48), highlighting that the chirality inversion of the peptide is the driving force for DOX release rather than heat-induced dissociation. While $PEG_2$-Kaz and $PEG_2$ should invert at 37 °C to efficiently release the DOX, the $PEG_2$-Cys provides excellent storage capabilities at physiological temperature for triggered release. To further probe if the DOX release was due to the physical inversion or differing affinities for M-type and P-type helices, we tested assembling the peptide with a pre-heated solution in an attempt to bypass the M-type helices and directly form the P-type structures. Structures assembled at 60 °C showed less than half the DOX incorporated compared to structures assembled at 4 °C (Supplementary Fig. 49). Additionally, DOX added to pre-assembled structures showed less binding although followed a similar trend where DOX had a higher affinity for M-type helices as compared to P-type helices.

Next, we explored the ability of our peptides to release DOX to cancer cells in culture and elicit differing killing responses based on their different loading efficiencies and pre-defined transition temperatures. To obtain doses that would allow us to observe differing responses in cell death rather than complete killing, all three peptides were assembled in the presence of 5 μM DOX rather than the maximal loading concentration (50 μM). The $PEG_2$ assemblies should be capable of delivering a final dose well above the IC50 if all DOX is released (Supplementary Figs. 50, 51), while $PEG_2$-Kaz, which has a lower loading capacity and $PEG_2$-Cys which has a similar loading but a transition temperature above 37 °C should be less impactful. After removing any DOX that was not intercalated in the structures, the structures were dried under vacuum to remove any HFIP before resuspending and adding them to cell cultures of both HeLa and MDA-MB-231 cells, two cancer cell lines that should respond to DOX treatment.

The initial effect of the drug-loaded materials on both cell types after 48 hours of incubation was assessed with live-dead (green and red, respectively) imaging (Fig. 7d). Compared to the no treatment control, a clear reduction in cell proliferation (fewer green cells) and indication of cell death (red cells) was observed after all three peptide treatments (Fig. 7d), with the greatest impact observed with the $PEG_2$ derivative as we expected. To quantify the differences between peptide treatments, an ATP-based cell viability assay was performed, which assesses the metabolic state of the cell, giving a more robust measurement of cell viability and a linear correlation between signal and cell number (Fig. 7e and Supplementary Fig. 52)[67]. In both cell types, significant reduction in viability for all DOX-loaded peptides compared to the control was observed after 48 hours, with the most

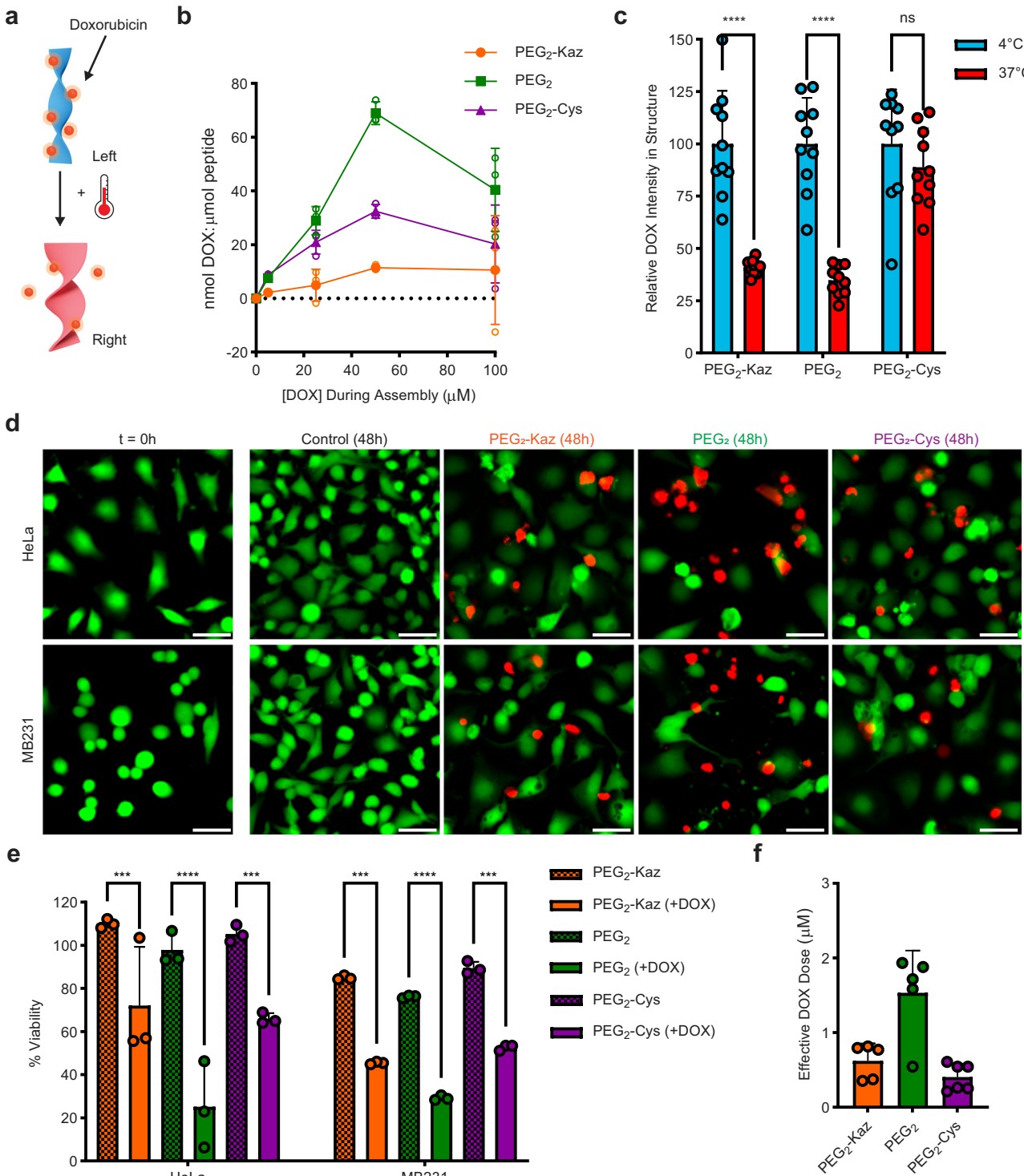

**Fig. 7 | Temperature-induced DOX release.** Schematic (**a**) of DOX incorporation and release. Quantification of DOX loading (**b**), open circles denote individual data points, solid symbols/error bars denote mean ± SD (*n* = 3 measurements of DOX loading from one bulk assembly). Quantification of DOX from confocal imaging (**c**) (*n* = 10 images, significance determined by a 2-way ANOVA with Sidak's multiple comparison test; left to right p values are 0.00000002, 0.000000001, 0.486). Live-dead assay images (**d**), green: live cells, red: dead cells (50 μm scale bars). Cell viability at 48 hours (**e**) normalized to untreated cells (*n* = 3 wells of cell culture, significance determined by a 2-way ANOVA with Tukey's multiple comparisons test; left to right p values are 0.00052, 0.0000000035, 0.00032, 0.00031, 0.000020, 0.00076). Effective DOX dose (**f**) after 48 hours (*n* = 5 cell assays for Kaz and PEG$_2$ samples, *n* = 6 for Cys). All error bars represent mean ± SD.

significant killing from the PEG$_2$ derivative, suggesting that it had successfully released a high level of DOX. While the PEG$_2$-Kaz peptide should also change conformation and release the DOX payload, the observed killing was less significant compared to the PEG$_2$ treatment. This difference can be explained by the four-fold lower loading

efficiency at the DOX concentration used. The DOX-loaded PEG$_2$-Cys peptide (high inversion temp) also showed a low degree of killing at a level similar to the PEG$_2$-Kaz, despite having high drug loading comparable to the PEG$_2$. Importantly, the peptides themselves showed minimal toxicity (Supplementary Fig. 53), indicating that the

differences observed were the result of varied amounts of DOX being delivered to the cells by each peptide.

The viability results reveal the effective dose of DOX released (Fig. 7f) and allow us to express it as a percentage of the maximal payload (Supplementary Fig. 54). Roughly, more than 70% of the DOX in PEG$_2$ and PEG$_2$-Kaz had been released while less than 25% of the DOX had been released from PEG$_2$-Cys, relatively consistent with the direct measurements of DOX release. Comparing the calculated dose and percent release between 24 and 48 hours for each peptide showed no significant differences (Supplementary Fig. 54), further supporting our hypothesis that DOX is retained stably within these structures until the transition temperature is reached, at which point the drug is released. Our ability to fine-tune the loading and release profiles of these biomaterials opens potential for combination therapies, where a mixture of peptides could be used to deliver an initial dose as well as enable long-term storage for triggered release as needed through local heating.

## Discussion

We have characterized the core peptide associated with Alzheimer's disease to uncover design rules for the supramolecular chirality of self-assembling amyloids. We provide insights into tuning the energy required for transitions between left- and right-handed β-peptide axial chirality through systematic peptide sequence modifications. The ability to enable inversion through small modifications to amyloid termini suggests that potential treatment routes could modify amyloid peptides to trigger chirality inversion and subsequent degradation of harmful amyloid plaques.

Using the guiding principles uncovered by our mutations to the core Alzheimer's associated peptide sequence, we designed a series of self-assembling peptides adopting chiral supramolecular structures that are able to shift from left- to right-handed helices by a tunable temperature switch. Interestingly, a parallel arises between our peptide library correlating faster nucleation rates to higher chirality inversion temperatures, and previous studies where chiral mutants (L to D amino acids) or conformational restriction was used to study the intermediate oligomeric states of amyloids (hypothesized to be more cytotoxic than the filamentous structures)[68,69]. In previous work by the Raskatov group, mutation of the E22 amino acid in Aβ42 from L to D chirality enabled stabilization of a pre-fibrillar form with increased cytotoxicity[70]. The use of a chiral mutant library to stabilize more non-fibrillar conformations of Aβ42 revealed that not all intermediate states exhibited increased cytotoxicity[71,72]. Although treating Alzheimer's disease was not the focus of our work, it is possible that the design rules we have uncovered could be utilized to design modifications that could promote rapid fibrillization of amyloids, thereby bypassing potential cytotoxic intermediates.

We successfully used our designer peptides to host anticancer therapeutic payloads and control their dosed-release by a temperature-triggered helical inversion. This peptide material is biocompatible in cell culture and able to be degraded upon releasing its cargo. These results highlight a platform for drug delivery based on a thermally actuated chirality inversion, where potential combinations of rapid and slow-releasing peptides could provide both short-term and long-term release with predefined dosing. While this is a promising approach, use of amyloid-like assemblies as drug delivery vehicles will need to be validated in vivo to test for potential cytotoxic effects[73]. In several previously published examples, amyloid-like assemblies have been utilized for drug delivery without major systemic toxicity[74–76]. For example, an enzyme-triggered diphenylalanine-based peptide assembly was recently shown to inhibit bone tumors in an orthotopic osteosarcoma mouse model with no observed systemic toxicity compared to a saline treatment[76]. Yet, as it is possible that off-target toxicity of amyloid drug delivery vehicles would be dependent on the tumor type and location within the body, further work with a variety of tumor models would be needed to establish what tumors may be safely treatable with this approach.

In addition to drug delivery, the physiologically relevant range of temperatures at which these peptides invert holds promise for biomedical applications. In addition, our data suggests that other inversion temperatures (outside of the physiologically relevant range) may be accessible through additional sequence modifications. Overall, these designer switchable chiral materials hold great promise and can be applied in other fields for applications such as enantioselective sensing or chiral-specific catalysis.

## Methods

All purchased reagents were used without further purification. Rink amide MBHA resin, Fmoc-Lys(azide)-OH ( > 99%), DIC (99%), and Fmoc-protected amino acids (>98%) were purchased from Chem-Impex. Fmoc-(PEG)$_2$-OH (97%) was purchased from PurePEG. Thioanisole (99%) was purchased from TCl. Oxyma pure (>99%), DMF (99%), ACN (99%), acetic anhydride (>98%), HFIP ( > 99%), α-chymotrypsin (>85%), fluorescamine (>98%), and doxorubicin·HCl (>97%) were purchased from Sigma Aldrich. TFA (99%), TIPS (98%), and ethanedithiole (>98%) were purchased from Fisher Scientific. Piperidine (99%), and DMSO ( > 99%) was purchased from VWR.

### Peptide synthesis and purification

Peptides were synthesized using an automated standard Fmoc solid-phase synthesis method (Liberty Blue, CEM) on Rink amide resin (100–200 mesh). For acetylated peptides, the N-terminus was acetylated with 20% acetic anhydride in DMF. Peptides were washed with DCM and cleaved from the resin with either 90% TFA, 5% thioanisole, 3% ethanedithiol, and 2% anisole solution or 95% trifluoroacetic acid (TFA), 2.5% triisopropylsilane (TIPS), and 2.5% dH$_2$O followed by filtration to remove the resin and evaporation of the acid.

Crude peptides were dissolved in acetonitrile (ACN) and water (1:1) and purified using either a Biotage Selekt or a Shimadzu reverse-phase HPLC with a C18 column and a gradient of water to ACN with 0.1% TFA. Peptide identity was confirmed by electrospray ionization mass spectrometry (Thermo Scientific LTQ-XL, LTQ Tune Plus 2.7 software) and purity was assessed using an analytical HPLC (Shimadzu UFLC, Lab Solutions 5.92). Peptides were lyophilized and stored at -80 °C.

If peptides were insoluble in the ACN:water mixture, the peptides were dissolved in ACN and precipitated with the addition of water. After three times of precipitation of the peptides in ACN and water, these purified peptides were ready for use and characterization.

### Peptide assembly

**Acetonitrile prep.** Purified peptides were dissolved at a final concentration of 1.3 mM in acidic conditions in 40% acetonitrile and 0.1% TFA (room temperature). Peptides were incubated for desired assembly times.

**HFIP Prep.** Peptides were dissolved at 2.5 mM in hexafluoroisopropanol (HFIP). The peptide solution was then added to sterile water at a 1:9 ratio to yield a final solution containing 250 μM peptide in 10% HFIP. Solutions were then either annealed (heated to 95 °C for 30 min and cooled to 25 °C at 1 °C/min) or aged 2 h at room temperature. Assemblies were stored at 4 °C.

### NMR

Isotopically $^{13}$C and $^{15}$N enriched peptides were synthesized using standard Fmoc solid-phase synthesis and purified by HPLC. After assembly, sodium sulfate was added to the mature sample and pelleted at 3000 x G[42]. The pellet was frozen in liquid nitrogen and the solvent was removed by lyophilization before measurements.

All experiments were performed on a Bruker (Billerica, MA) Avance 600 spectrometer using a Bruker 4 mm HCN biosolids magic-angle spinning (MAS) probe. The MAS frequency was maintained at 10 kHz ± 2 kHz. To ensure no degradation occurred by MAS or RF heating during the experiment, cooling air was used to maintain sample temperature below -1°C. $^{13}$C (150.9 MHz) CP-MAS spectra were taken before and after $^{13}$C{$^{15}$N} REDOR experiments[49,50] to ensure no change occurred in the sample. To compensate for pulse imperfections, xy8 phase cycling[77] of $^{13}$C{$^{15}$N} REDOR 4 and 8μs rotor-synchronized $^{13}$C and $^{15}$N π-pulses, respectively, and EXORCYCLE phase cycling[78,79] of the final $^{13}$C Hahn-echo refocusing pulse is applied with 128 kHz Spinal 64[80] $^1$H (600.133 MHz) decoupling. $^{13}$C{$^{15}$N} REDOR data points were calculated as the sum of the center and sideband integrated peak heights.

## Small angle X-ray scattering
Peptides were assembled at 1.3 mM in 40% acetonitrile, 0.1% TFA at room temperature for 2 weeks before SAXS measurements were taken at Argonne National Laboratory. Measurements were taken at room temperature at APS using the 12-ID beam-line. A 15 cm × 15 cm, high-resolution, position-sensitive, nine element-tiled, CCD mosaic detector was used to collect the data. About five measurements were recorded per sample and the data was averaged to create a scatter profile (0.5 s exposure, 2 m detector distance, 12 keV). A Biologic SFM 400 stopped-flow apparatus with a cylindrical quartz capillary (1.0 mm diameter, 0.01–0.02 mm wall thickness) was used to continuously flow 100 μL of peptide solution into the X-ray beam. Solution measurements were taken to ensure cleanliness of the sample cell between sample measurements and to obtain values for background subtraction.

## X-ray diffraction
Mature peptide nanotubes were bundled with 20 mM sulfate or magnesium at a peptide to salt ratio of 1:10. The resulting white precipitate was collected by centrifugation. The pellet was frozen and lyophilized to yield dry powder for XRD. Powder spectra were obtained using a Bruker D8 Discover diffractometer equipped with a multi-position X, Y, Z stage, a cobalt/copper X-ray tube with Goebel mirror, and a Vantec-1 solid-state detector. The sample was placed in a zero-background holder on the stage and the spectrum obtained using Bragg-Brentano geometry. The scan step was repeated several times to maximize the diffracted intensity and minimize noise.

## Transmission electron microscopy
TEM grids were prepared by incubating 10 μL of sample on a 300-mesh copper-carbon film grid for 5-10 minutes (1 min for nanotubes). Excess solution was wicked off and the grids were rinsed twice with sterile water. Grids were then stained twice with 2% aqueous uranyl acetate, 20 seconds each (or once for 1 minute for nanotube samples). Grids were imaged on a Tecnai T12 at 120 kV (Digital Micrograph 3 software). Fiji ImageJ 1.54 h software (NIH) was used for quantification and to make minor adjustments to the images for brightness and contrast.

For TEM experiments to follow structure transition, assembled peptide solution was heated to each temperature and allowed to equilibrate 10 minutes before an aliquot of the solution was transferred to a TEM grid (at room temperature).

Electron tomography was acquired on a Tecnai T12 at 120 kV equipped with SerialEM 4.1 and images were processed using IMOD 4.11.24 software.

## Electron diffraction
Electron diffraction samples were prepared by first diluting the nanotube samples to 0.65 mM in 40% ACN, 0.1% TFA. 15 μL of diluted sample was incubated on a 300-mesh carbon-copper grid for 1 minute. Excess sample was wicked from the grid from one side using filter paper. Unstained grids were dried under vacuum for 5 minutes. Diffraction patterns were collected on a Talos F200X TEM at 200 kV.

## Circular dichroism
Samples were assembled as described in the methods for peptide assembly before being measured on CD. Peptide solutions were individually diluted as needed (up to 5-fold) to obtain a clear signal without saturating the detector. Spectra were collected with a 1 mm path length cuvette on a Chirascan Plus (or Chirascan V100) CD spectrophotometer (Chirascan 4 software). For melting curves, samples were heated at 1 °C/min. For nucleation rates, all peptides were assembled at 50 μM and CD at 216 nm were collected immediately after the peptide solution in HFIP was diluted into water.

## Scanning electron microscopy
For SEM, 7.5 μL of samples were dried on silicon wafers and sputter coated with 4 nm gold. Samples were then imaged on an FEI Helios 600 Dual Beam System SEM.

## FT-IR spectroscopy
10 μL aliquots of peptide solution were dried as thin films on a ATR diamond crystal. Spectra was obtained at room temperature averaging 500 to 800 scans of 2 cm$^{-1}$ resolution with 5 mm aperture and 4 mm/sec scanning speed, using MCT and TGS detectors. Spectra were processed with zero-filling and a cosine apodization function. IE-IR spectra were normalized to the peak height of the $^{12}$C band.

For Fmoc-FFFF-PEG$_2$, samples were prepared by drying 10 μL of sample onto a horizontal attenuated total reflectance (ATR) ZnSe crystal (PerkinElmer FTIR spectrometer). Measurements were collected in absorbance mode between 1800 - 1500 cm$^{-1}$ at a resolution of 1 cm$^{-1}$ and averaged over 32 scans. Background, solvent blank and baseline corrections were performed on the PerkinElmer instrument software.

## Confocal microscopy
For FRAP experiments and movies of heating, Fmoc-FFFF-PEG$_2$ peptide assemblies were stained with 10 mol% thioflavin T (Sigma, >65%). Samples were diluted to 125 μM in 10% HFIP and 20% glycerol (Sigma, >99%) was added. Movies were collected on a Zeiss 880 confocal laser-scanning microscope with Airyscan using a 458 nm excitation laser, Plan-Apochromat 63X/1.4 oil objective, and an incubated enclosure (software: Zen 2.3 SP1 FP3 (black)). For FRAP, the sample was allowed to equilibrate 10 min at each temperature before movies were recorded.

## Chymotrypsin degradation assays
For chymotrypsin degradation assays, α-chymotrypsin was reconstituted at 50 μg/mL in 1 mM HCl with 2 mM CaCl$_2$ (Sigma, >93%). The final reaction solution contained 50 μM pre-assembled peptide and 5 μg/mL chymotrypsin in 100 mM Tricine (Sigma, >99%), 20 mM CaCl$_2$ pH 7.8. The reactions were incubated at 30 °C and aliquots were taken at over 48 h.

To quantify degradation, a fluorescamine assay was used to measure the concentration of free amines produced as the peptide is digested. Briefly, each 100 μL aliquot was mixed with 33.3 μL fluorescamine (10.8 mM in DMSO) and shaken at room temperature for 5 min. A calibration curve for concentration of amines in the assay was generated using dilutions of glycine measured under the same conditions. Fluorescent measurements were taken on a Perkin Elmer Enspire plate reader (excitation: 387 nm, emission: 480 nm). For product analysis, reactions were filtered through a 3KDa filter to remove protein before analytical HPLC (Shimadzu UFLC) and MS (Thermo Scientific LTQ-XL) were measured.

To measure remaining activity, chymotrypsin (200 μg/mL) was used to digest left-handed (aged) and right-handed (annealed) Fmoc-

FFFF-PEG$_2$ (10 μM) for 24 h at 30 °C. After 24 h, the remaining activity of enzyme was measured by the addition of 100 μM $N$-succinyl-ʟ-pheny-lalanine-p-nitroanilide (Sigma, 98%) and absorbance was monitored at 410 nm.

### Doxorubicin incorporation and release
**Loading assays.** DOX-HCl was solubilized in HFIP. This solution was then used to solubilize the peptides which were then assembled following the previous protocol with aging carried out on ice to prevent any structure transition for the PEG$_2$-Kaz peptide. The peptide solutions were centrifuged at 21,382 x g for 30 min at 4 °C to precipitate any structures. The supernatant was removed and quantified from fluorescence (excitation: 488 nm, emission: 596 nm) using a Thermo Scientific Varioskan LUX plate reader (SkanIT 6.1 software).

**Release assays.** DOX-HCl was solubilized at 500 μM in HFIP. This solution was used to solubilize the peptides which were then assembled following the previous protocol with aging carried out on ice. Aliquots of DOX-loaded peptides were then heated to 37 °C for 1 h. Samples were then centrifuged at 21,382×$g$ for 30 min at 4 °C and the supernatant containing released DOX was removed. The remaining assemblies in the pellet were resuspended in 10% HFIP supplemented with 20% glycerol for CLSM imaging on a Zeiss LSM 710 (Zen Black 2.3 SP1 FP3) using a 458 nm excitation laser with emission collected from 535-650 nm (Plan-Apochromat 40X/1.4 Oil objective). Images were collected as 15 μm z-stacks with 0.2 μm intervals. ImageJ was used for all CLSM quantification. A max intensity z-projection was performed on all images followed by an auto-threshold to select only areas with material present. The mean gray value of the selected area was then used as a relative measurement of DOX remaining in the material.

**Cell viability assays.** DOX-HCl was solubilized at 50 μM in HFIP and this solution was used to solubilize the peptides which were then assembled following the previous protocol with aging carried out on ice. Assemblies were then centrifuged at 21,382×$g$ for 30 min at 4 °C. The supernatant was removed, and the pellets were dried under vacuum to remove any residual HFIP. The structures were resuspended in cell culture media before adding to cells.

### Cell culture
MDA-MB-231 cells (ATCC HTB-26), derived from a 51-year-old Caucasian female, were obtained from ATCC. HeLa cells, a cervical adenocarcinoma cell line (ATCC CCL-2), derived from a 31-year-old Black female, were obtained from the UNC Tissue Culture Facility. Both cell lines were cultured in DMEM, High Glucose (Gibco, 11965092) supplemented with 10% Fetal Bovine Serum (Seradigm, 1500-500 G) and 1% Penicillin-Streptomycin (Gibco, 15140122). Cells were maintained at 37 °C, 5% CO$_2$ and passaged every 3 days according to ATCC guidelines.

**Cell viability assays.** For all cell viability assays, cells were plated in opaque 96-well plates at a seeding density of $3 \times 10^4$ cells/cm$^2$. Cells were plated in full cell culture media (DMEM, 10% FBS, 1% P/S) and allowed to adhere to the plate for 12 hours prior to the addition of doxorubicin or doxorubicin-containing assemblies.

**Doxorubicin dose response curves.** For doxorubicin dose response curves, a 50 mM stock solution of DOX-HCl in molecular biology grade water was prepared immediately prior to the start of the assay. This stock solution was used to prepare Doxorubicin standards in full cell culture media at concentrations ranging from 3 nM to 50 μM. After cells were allowed to adhere to the plate, the culture media was removed and replaced with doxorubicin-containing media or a no-drug control. All standards were run in triplicate, and cell viability was measured at 24 and 48 hours using the CellTiter-Glo 2.0 Cell Viability

Assay (Promega, G9241) according to the manufacturer's instructions. Luminescence values were measured using a Biotek Synergy HTX Multi-Mode Reader in luminescence mode, with a 1 mm read height and a 1 second integration time.

**Killing assays with DOX-loaded structures.** DOX-Loaded structures were prepared as described in the methods for doxorubicin incorporation and release. Immediately prior to the start of the assay, dried structures were resuspended in the cell culture media to a final peptide concentration of 250 μM. After the cells were allowed to adhere to the plate, the culture media was removed and replaced with the solutions of DOX-loaded assemblies made from each of the 3 peptide derivatives. Each peptide assembly was run in triplicate. Cell viability was measured at 24 and 48 hours using the CellTiter-Glo 2.0 Cell Viability Assay using the method detailed for the DOX dose-response curves.

**Live/Dead Assays.** In addition to the continuous method of quantifying cell viability, live/dead assays using Calcein AM and Propidium Iodide were performed on cells incubated with DOX-loaded structures for 48 hours. After incubating the cells with the assemblies, the cell culture media was removed and replaced with PBS containing 2 μM Calcein AM (Invitrogen, >95%) and 3 μM Propidium iodide (Sigma, 94%). Cells were labeled for 30 minutes at room temperature followed by imaging using a GE INCell Analyzer 2200 high-content microscope at 10x using FITC and Texas Red filter sets (IN Cell Analyzer 2200 software, version 7.2).

### Cell data analysis
**Dose response curve fitting.** All data analysis was performed in GraphPad Prism 9. Raw luminescence values from DOX dose response assays were first normalized to the mean of the no-drug control wells. This value was set to 100% viability. The data was then fit using the built-in Prism non-linear fit [Inhibitor] vs Normalized Response – Variable Slope using a standard least-squares regression. All replicates were used for the fitting and data points came from 3 independent experiments. Equations from the curve fits were used to calculate effective doses from cell viability results with peptides. The percent release was calculated by comparing the effective dose to the total incorporated DOX.

**ImageJ analysis.** Live/dead assay images were processed using Fiji/ImageJ, v1.53 s (National Institute of Health). Low magnification tiled fluorescent images were loaded into Fiji and first stitched into one final image for each well using the Grid/Collection stitching plug-in. Representative regions were then selected for illustration in the figures.

### Statistics and reproducibility
All statistical analysis was performed in GraphPad Prism (versions 9 and 10). All error bars shown represent the standard deviation of the data points from the mean unless otherwise noted. The n values and the nature of each n for each experiment are detailed in each figure caption. Replicates were taken from unique samples except for confocal imaging where replicates were taken in unique areas of the same sample. For ANOVA results, multiple comparisons were made using Tukey's multiple comparisons test, Sidak's multiple comparison test or the two-stage linear procedure of Benjamini, Krieger, and Yekutieli. All $t$-tests performed were unpaired, two-tailed $t$-tests. Number of asterisks for p values are as follows: *$p < 0.05$, **$p < 0.01$, ***$p < 0.001$, ****$p < 0.0001$. Degrees of freedom = $n$-1 for $t$ tests. Additional details on each statistical test are provided in the source data file.

All attempts to reproduce the data in this manuscript were successful. The number of trials for the individual experiments are specified in the figure legends.

## Inclusion and ethics statement

All relevant researchers that participated in the development and data collection of the study were included in the author list. The research in the manuscript is not region-specific; relevant research was cited regardless of the location in which it occurred.

## Reporting summary

Further information on research design is available in the Nature Portfolio Reporting Summary linked to this article.

## Data availability

The spectral and imaging data generated in this study are available in the manuscript and supporting files. Data underlying plots is available in the source data file supplied with the manuscript. If other formats of the data generated during the current study are needed, they are available from the corresponding author upon request. Source data are provided with this paper.

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

## Acknowledgements

This work was supported by the University of North Carolina at Chapel Hill (UNC-CH) and Emory University. Mass spectrometry was performed at the UNC Mass Spectrometry Core Laboratory supported in part by the University of North Carolina's School of Medicine Office of Research. Transmission electron microscopy, FRAP, heating videos, and fluorescent cell images were taken at UNC Hooker Imaging Core Facility, supported in part by P30CA016086 Cancer Center Core Support Grant to the UNC Lineberger Comprehensive Cancer Center. Imaging studies were also supported by Robert P. Apkarian Integrated Electron Microscopy Core (IEMC) at Emory University, which is subsidized by the School of Medicine and Emory College of Arts and Sciences. Additional support for IEMC was provided by the Georgia Clinical and Translational Science Alliance of the National Institutes of Health under award number UL1TR000454. Circular Dichroism was performed at UNC Macromolecular Interactions Facility supported by the National Cancer Institute of the National Institutes of Health under award number

P30CA016086. Confocal microscopy of DOX structures was performed at UNC Microscopy Services Laboratory, supported in part by P30CA016086 Cancer Center Core Support Grant to the UNC Lineberger Comprehensive Cancer Center. SEM and electron diffraction was performed at the Chapel Hill Analytical and Nanofabrication Laboratory, CHANL, a member of the North Carolina Research Triangle Nanotechnology Network, RTNN, which is supported by the National Science Foundation, Grant ECCS-2025064, as part of the National Nanotechnology Coordinated Infrastructure, NNCI. R.F. and S.J.K. acknowledge financial support from the Alfred P. Sloan Foundation grant G-2021-14197 (R.F.). R.F. also acknowledges additional support from the Cottrell Scholar Award #CS-CSA-2023-033 (R.F.) sponsored by Research Corporation for Science Advancement. We are further grateful for support from NSF DMR-2004846 BMAT (D.G.L.) in collaboration with BSF 2019745 (D.G.L.) for some of the peptide synthesis resources accessed from NIH Alzheimer's Disease Research Center P50AG025688. The content is solely the responsibility of the authors and does not necessarily reflect the official views of the National Institutes of Health. We thank John Bacsa for the powder XRD which was performed by the X-ray Crystallography Center at Emory University. We are grateful to Jeannette Taylor and Hong Yi in the Emory Robert P. Apkarian Microscopy Core for TEM advice and training. We also acknowledge the Division of Chemical Sciences, Geosciences, and Biosciences, Office of Basic Energy Sciences of the U.S. Department of Energy through Grant DE-FG02-02ER15377 (D.G.L.) for some of the peptide synthesis and analyses, and initial support from the NSF and NASA Astrobiology Program, under the NSF Center for Chemical Evolution, CHE-1004570. We thank Matthew Sheldon, and Justin Hill from the Freeman team, for help with the graphic illustrations included in the manuscript.

## Author contributions

The manuscript was written with contributions and approval from all authors. S.J.K. characterized all peptides with the assistance of co-authors, designed and executed all experiments for phenylalanine-based peptides, and assisted with writing the manuscript. M.L. assisted with synthesis and characterization of KLVFFA(E), (L), and (V) peptides and assisted with drafting the manuscript. T.J. and Q.W. contributed to peptide synthesis and purification. K.D.R. and S.B. performed cell treatments and analysis and assisted with writing. Y.G. assisted with electron diffraction of nanotubes and SEM imaging. M.L.D. performed FTIR analysis of Fmoc-FFFF-PEG2 peptide. W.S.C., T.O.O., and A.K.M. contributed to the initial characterization of nanotubes with CD, SAXS, NMR, and TEM. D.G.L. and R.F. supervised the research, procured funding, and assisted with writing the manuscript.

## Competing interests

The authors declare no competing interests.
