## [Peer Review File · Nature Communications]

Uncovering Supramolecular Chirality Codes for the Design of Tunable BiomaterialsREVIEWER COMMENTS

Reviewer #1 (Remarks to the Author):

This paper investigates the morphological transformations of B-amyloid based peptides. The role of amino acids on the organization is very well explained in it. Interestingly, morphology engineering is an emerging term and explained in early 2016.

This paper deals with the intricacies and the finer details of right and left helical supramolecular assembly. The studies are very well done with all the experimental details.

The following are my queries:

The authors argue about the resistant nature (Chymotrypsin digestion) of the left handed supramolecular helix of the peptide (Figure 6). Careful analysis of the Fig. 6 revealed that within 10 hrs the left handed one undergo faster kinetics compared to the right. Only the final one point is making the difference and the conclusion by the authors is mainly based on this last data point. The faster kinetics in 10 h favors the opposite argument.

Alternately, one of the product from the cleavage of left handed helical structure is the Fmoc unit (probably an alkene). This could covalently modify the side chain of Ser of chymotrypsin, thereby inactivating the enzyme. The observed decrease after 10 h then may not arise from the morphological difference, but rather due to inhibition of enzyme by one of the products ie the alkene.

The other point is about the helicity. It is good to mention how the helicity is decided, since there are different viewpoints on the supramolecular helical chirality determination using CD data. A brief description on the method adopted will be useful to the readers and give clarity. If BeStSel is used in the quantification, then adding that in the supporting data will be useful.

In Figure 2F, there is no cross over in the left handed helix (85 oC).

The release of the DOX-drug, making use of the left to right transition is an interesting aspect of this work. The added advantage is the biocompatibility of the peptides.

Is the affinity of DOX higher for left handed helical assembly compared to right? The temperature is definitely a cause for DOX dissociation, but after the disassembly it can still bind to the right handed assembly (Figure 7A).

The work is of a high quality and I support the publication of this work.

Reviewer #2 (Remarks to the Author):

The authors report findings that should be of interest to the field of Amyloid research, but I am not convinced that the study merits publication at Nat. Commun. level. Sci. Rep. may be more appropriate.

The authors should perform additional cell culture experiments using cell lines that are more common in the field of AD, such as PC12 and/or SH-SY5Y. The authors may also wish to perform additional experiments in primary neurons to support their key findings, but this is optional as I realize that working with primary neurons is time-consuming and potentially quite demanding.

The discussion is very short and has to be expanded. There is a number of papers that studied chiral effects on Abeta aggregation and none of them are mentioned. The authors should include the following papers in their discussion to provide a more balanced perspective:

- (a) Curr. Opin. Chem. Biol. 2021, 64, 1-9.
- (b) J. Org. Chem., 2020, 85, 1385-1391.
- (c) ACS Chem. Neurosci., 2019, 10, 3880-3887.
- (d) Sci. Rep. 2017, 7:12433.
- (e) Chem. Eur. J. 2016, 22, 11967-11970

Reviewer #3 (Remarks to the Author):

In this work, the authors designed a series of peptides and achieved structural transformation of self-assembled nanostructures through a heating process, which are further utilized for drug loading and release. Many experimental approaches such as TEM, CD, and SAXS have been adopted to characterize the structural transformation and provided valuable information. However, the current data are still not sufficient to prove that the self-assembled nanostructure has transformed from a left-handed manner to a right-handed manner after heating. Before this issue can be proven by further experiment, the title of this manuscript would be somewhat questionable.

1. It is necessary to carefully characterize the left-handed and right-handed nanostructures generated in this work through AFM and SEM to confirm this point.

2. This manuscript mainly explains the transformation from left-handed to right-handed through the changes in CD signal. However, CD data can only indicate that the conformation of molecular monomers has undergone a transformation, but it is not sufficient to indicate that the chirality of self-assembled twisted nanostructures has undergone a reversal. Because many self-assembly experiences with negative cotton signals exhibit right-handed twist, while self-assembly experiences with positive cotton signals exhibit left-handed twist.

3. Figure S31 shows the AFM scans of Fmoc-4F-PEG2 aged (a) or annealed (b). The AFM images of Fmoc-4F-PEG2 aged and annealed look similar. It seems that both are right-handed. The AFM data is recommended to be re-analyzed carefully. More structural details of the supramolecular nanostructures are needed.

4. Due to the possibility of placing the copper mesh in the opposite (mirror) direction during characterization, it requires extra-caution when used TEM to determine whether left-handed or right-handed helices. Therefore, SEM experiments are recommended.
5. Before and after heating, detailed characterization and analysis of the arrangement and conformational changes of molecules are required, and it is necessary to demonstrate that the changes caused by the arrangement or conformational changes of molecules are sufficient to alter the supramolecular chiral structure through chiral transfer.
6. Solid-state NMR ^{13}C - ^{15}N distance measurements have been adopted to confirm the out-of-register antiparallel β -sheets before heating. Would the NMR ^{13}C - ^{15}N distance change after heating (the structural transformation)?
7. Following the question above, how did the alignments between peptides and molecular conformation of a peptide change during the transformation?
8. Another minor errors: The wavelength labels in the Figure 1D are reversed.

Reviewer 1

This paper investigates the morphological transformations of B-amyloid based peptides. The role of amino acids on the organization is very well explained in it. Interestingly, morphology engineering is an emerging term and explained in early 2016. This paper deals with the intricacies and the finer details of right and left helical supramolecular assembly. The studies are very well done with all the experimental details.

We thank the reviewer for the positive assessment of our manuscript. We appreciate the use of the emergent term “morphology engineering” providing context to the scope of the work.

1. The authors argue about the resistant nature (Chymotrypsin digestion) of the left-handed supramolecular helix of the peptide (Figure 6). Careful analysis of Fig. 6 revealed that within 10 hrs the left-handed one undergoes faster kinetics compared to the right. Only the final one point is making the difference and the conclusion by the authors is mainly based on this last data point. The faster kinetics in 10 h favors the opposite argument. Alternately, one of the products from the cleavage of left-handed helical structure is the Fmoc unit (probably an alkene). This could covalently modify the side chain of Ser of chymotrypsin, thereby inactivating the enzyme. The observed decrease after 10 h then may not arise from the morphological difference, but rather due to inhibition of enzyme by one of the products ie the alkene.

The reviewer is correct that the initial time points had shown slightly faster degradation of the left-handed assemblies. To get a clearer view of the degradation kinetics, we adopted a revised fluorescamine protocol where the fluorescamine is dissolved in DMSO (instead of 25% acetone) which improves the solubility of the fluorescamine and the self-assembled structures allowing for more complete quantification of the degradation. We repeated the assay and took time points out to 48 hours. This revised assay (Fig. 6b, S41) clearly shows faster degradation of the right-handed assembly at all time points. Furthermore, the TEM (Fig. 6d) supports that the right-handed assemblies are more degraded than the left-handed assemblies.

We also probed for the possibility, suggested by the reviewer, that a byproduct may inactivate the enzyme. We used a colorimetric substrate to assess the activity of chymotrypsin after degrading either left- or right-handed structures and showed that there are no significant changes to the enzyme activity (Fig. S46). Discussion was added on page 14, lines 7-12.

2. The other point is about helicity. It is good to mention how the helicity is decided, since there are different viewpoints on the supramolecular helical chirality determination using CD data. A brief description of the method adopted will be useful to the readers and give clarity. If BeStSel is used in the quantification, then adding that in the supporting data will be useful.

We thank the reviewer for bringing up this point. We have added a section of supplemental text describing how chirality is defined on both the β -sheet level and the supramolecular ribbon level (See SI Note 1, referred to on page 5, lines 6-7 in the main text). BeStSel was not used for quantitative analysis of the chirality, but we adopted their definition of left- and right-handed β -sheet axial chirality based on the sign of the ~ 219 nm peak which we now describe in the text and cite the appropriate references. Additionally, to help differentiate helicity of the β -sheets and the helicity of the supramolecular ribbon (which was determined by imaging), we have made use of the terms M-type and P-type helices for referring to the handedness of the ribbons while we retain left-handed and right-handed terminology when describing β -sheet twists.

3. In Figure 2F, there is no crossover in the left handed helix (85 oC).
This is correct, the KLVFFAV peptide remains left-handed at 85°C, and transitions to right-handed during the slow annealing process. To avoid confusion, we have moved this plot showing the cooling to the SI (Fig. S13). We show the aged (left-handed) vs. annealed (right-handed) states of KLVFFAV in the current Fig. 2d and e. We specify this in the text on page 7, lines 21-24.
4. The release of the DOX-drug, making use of the left to right transition is an interesting aspect of this work. The added advantage is the biocompatibility of the peptides. Is the affinity of DOX higher for left-handed helical assembly compared to right? The temperature is definitely a cause for DOX dissociation, but after the disassembly it can still bind to the right-handed assembly (Figure 7A).

We thank the reviewer for this important comment. To understand the affinity DOX has for the different structures, we have added several experiments. To evaluate if DOX was able to be incorporated into the right-handed structures during assembly, we assembled peptides with pre-heated solutions to bypass the left-handed state and showed that approximately 60% less DOX was incorporated in right-handed structures as compared to the left-handed structures. Furthermore, to show the importance of drug loading during assembly, we also added DOX to pre-assembled left-handed structures showing that 75% less DOX will bind if added to pre-assembled structures rather than being present during assembly; no DOX binding was observed when added to pre-assembled right-handed structures (Both results in Fig. S49). These experiments point toward DOX having higher affinity for the left-handed assembly. Additionally, while temperature may induce some dissociation of DOX from the structures, the structural reconfiguration is the primary cause for release as the cysteine modified peptide (left-handed at 37°C) does not release a significant amount of DOX simply by heating, unless the inversion temperature was met (Fig. 7c). See page 16, lines 11-21.

Reviewer 2

The authors report findings that should be of interest to the field of Amyloid research, but I am not convinced that the study merits publication at Nat.Comm. level. Sci.Rep. may be more appropriate.

We respectfully disagree. We believe our findings are of high interest to Nat. Commun readers of multiple research areas. Briefly, the ability to tune supramolecular chirality inversion is a challenge across all chiral supramolecular structures not only in amyloids. The tunability of our system makes it ideal as a biomaterial for drug delivery and other applications. As we demonstrated, the structural reconfiguration at body temperatures was used to effectively release an anticancer chemotherapy drug. Beyond drug delivery, as highlighted by several extensive reviews on supramolecular chirality, other applications are possible such as enantioselective sensing or catalysis (Shen *et al. Prog. Polym. Sci.* 123, 101469 (2021); Liu *et al. Chem Rev* 115, 15, 7304 (2015)). Moreover, other systems attempting to switch supramolecular chirality through other means have been reported in Nature Communications or similar levels of journals, suggesting interest of broad readership in this research area (Li *et al, Nat Commun* 14, 5030 (2023); Kim *et al, Nat Commun* 6, 6959 (2015); Wang *et al, Adv. Mater* 27, 2065 (2015))

1. The authors should perform additional cell culture experiments using cell lines that are more common in the field of AD, such as PC12 and/or SH-SY5Y. The authors may also wish to perform additional experiments in primary neurons to support their key findings, but this is optional as I realize that working with primary neurons is time-consuming and potentially quite demanding.

We believe this comment stems from a misunderstanding by this reviewer. While we utilize design rules from Alzheimer's Disease (AD) related peptides to engineer structures which can undergo chirality inversion, the application of the work is not intended to treat AD related cells. Rather, we are using the design rules we derived from the amyloid system to yield a reconfigurable drug-delivery vehicle able to release the anti-cancer drug doxorubicin. As such, we chose two cancer cells (HeLa and MB231) to illustrate the effectiveness of our therapy. We clarify this aspect on page 15, lines 8-10 and page 16, lines 33-36.

2. The discussion is very short and has to be expanded. There are a number of papers that studied chiral effects on Aβ aggregation and none of them are mentioned. The authors should include the following papers in their discussion to provide a more balanced perspective:

- (a) Curr. Opin. Chem. Biol. 2021, 64, 1-9.
- (b) J. Org. Chem., 2020, 85, 1385-1391.
- (c) ACS Chem. Neurosci., 2019, 10, 3880-3887.
- (d) Sci. Rep. 2017, 7:12433.
- (e) Chem. Eur. J. 2016, 22, 11967-11970

We appreciate the need for a more balanced perspective on chirality. While our original draft primarily focused on the switchable supramolecular chirality of a L-amino acids peptide sequence through sequence modifications, it is important to discuss the work done with D-amino acids to probe the effects of chirality. We have added the references suggested by the reviewer and discuss some of the similar trends which may be observed between the mutations we have designed, and chiral mutations discussed in these papers. Page 19, lines 16-27.

Reviewer 3

In this work, the authors designed a series of peptides and achieved structural transformation of self-assembled nanostructures through a heating process, which are further utilized for drug loading and release. Many experimental approaches such as TEM, CD, and SAXS have been adopted to characterize structural transformation and have provided valuable information. However, the current data are still not sufficient to prove that the self-assembled nanostructure has transformed from a left-handed manner to a right-handed manner after heating. Before this issue can be proven by further experiment, the title of this manuscript would be somewhat questionable.

We thank this reviewer for their careful review of this manuscript and the important comments. We have added additional experiments to prove the chiral transformation and added additional sections of text to address questions that were raised, as detailed below.

1. It is necessary to carefully characterize the left-handed and right-handed nanostructures generated in this work through AFM and SEM to confirm this point.
To confirm the chirality that we observed on TEM, we have added SEM imaging (Fig. S30) which matches our previous results, showing an inversion from left-handed (M-type) to right-handed (P-type) ribbons.
2. This manuscript mainly explains the transformation from left-handed to right-handed through the changes in CD signal. However, CD data can only indicate that the conformation of molecular monomers has undergone a transformation, but it is not sufficient to indicate that the chirality of self-assembled twisted nanostructures has undergone a reversal. Because many self-assembly

experiences with negative cotton signals exhibit right-handed twist, while self-assembly experiences with positive cotton signals exhibit left-handed twist.

We thank the reviewer for this comment. We agree that the CD signals describe the chirality of the β -sheet twist rather than the supramolecular chirality. We have added a section of supplemental text describing how we have defined chirality at each length scale (See SI Note 1) as we discuss both β -sheet chirality and supramolecular chirality of the ribbons in this manuscript. The CD was used to determine the β -sheet chirality, while imaging was used to determine supramolecular chirality. Per this reviewer suggestion, we confirmed the supramolecular chirality inversion via SEM (Fig. S30) as well as TEM showing that the helical inversion occur in temperatures where β -sheet inversion takes place (Fig 5a). To clarify the text, we have utilized the terms M-type and P-type helices to refer to left- and right-handed supramolecular structures, respectively. When referring to the β -sheet twists, we use the nomenclature of left- and right-handed β -sheets. This clarification is updated in the manuscript and in the supplemental text.

3. Figure S31 shows the AFM scans of Fmoc-4F-PEG2 aged (a) or annealed (b). The AFM images of Fmoc-4F-PEG2 aged and annealed look similar. It seems that both are right-handed. The AFM data is recommended to be re-analyzed carefully. More structural details of the supramolecular nanostructures are needed.

We agree with this reviewer that it is difficult to resolve the AFM images. To address this, we have removed the AFM scans in question, and instead added SEM characterization (Fig. S30) to show the supramolecular chirality. We also replaced the spectra from the AFM-IR with higher resolution FT-IR spectra (Fig. S36) taken from the bulk samples (See text on page 13, lines 2-6).

4. Due to the possibility of placing the copper mesh in the opposite (mirror) direction during characterization, it requires extra-caution when used TEM to determine whether left-handed or right-handed helices. Therefore, SEM experiments are recommended.

We appreciate the reviewer's comment concerning the risk of chirality assignments from TEM. We have added SEM images of the structures showing the left- and right-handed supramolecular chirality which matches the observations from TEM (Fig. S30). Of note, all TEM grids were placed with the copper mesh in the same orientation, so that TEM comparisons are consistent.

5. Before and after heating, detailed characterization and analysis of the arrangement and conformational changes of molecules are required, and it is necessary to demonstrate that the changes caused by the arrangement or conformational changes of molecules are sufficient to alter the supramolecular chiral structure through chiral transfer.

To address this reviewer comment, we have conducted the following:

- (a) To characterize the molecular conformation of the Fmoc-FFFF-PEG2 before and after heating, we have utilized both FTIR and CD spectra. FTIR spectra show an anti-parallel conformation of β -sheets both before and after heating, with the annealed sample having a higher frequency peak compared to the aged sample (Fig. S36). This is likely due to increased hydrophobic packing as the β -sheets laminate to form the larger right-handed structures (see text on page 13, lines 2-6). The CD spectra also showed β -sheet signals in both cases but

with opposite Cotton effects (negative to positive) showing that the chirality of the β -sheet was inverted (Fig. 4c).

(b) To demonstrate that the chirality change of the monomers can amplify the chirality of the supramolecular structure, we have designed the enantiomeric peptide (Fmoc-FFFF-PEG2 with all D amino acids; Fig. S20 and S21). CD spectra revealed a reversed trend on the β -sheet level going from right-handed to left-handed with heating. Using TEM and SEM imaging, we confirmed that the chirality of the supramolecular structure matched that of the β -sheet, going from small P-type ribbons (right-handed) to larger M-type ribbons (left-handed). As the enantiomeric peptide also demonstrates a reversed chirality with heating, we believe that this supports the notion that the chirality of the supramolecular structure is altered through chiral transfer. See text on page 8, lines 25-35.

6. Solid-state NMR ^{13}C - ^{15}N distance measurements have been adopted to confirm the out-of-register antiparallel β -sheets before heating. Would the NMR ^{13}C - ^{15}N distance change after heating (the structural transformation)?

This is an insightful question. Since aged KLVFFAL (left-handed) and aged KLVFFAV (right-handed) assemblies in 40% Acetonitrile exhibit almost identical NMR measurements corresponding with out-of-register β -sheets, it suggests that the chirality of the β -sheet is a parameter that does not shift the peptide registry to change the ^{13}C - ^{15}N distances. We have added text to address this comment on page 4 line 4 through page 5 line 12.

7. Following the question above, how did the alignments between peptides and molecular conformation of a peptide change during the transformation?

Based on the NMR measurements of KLVFFAE/KLVFFAL (left) and KLVFFAV (right), i.e. opposite handedness, we see they all have out of register sequences. Previous work by Favrin *et al.* (Ref. 51) showed that there is only one likely out-of-register alignment for this peptide sequence, suggesting that both the left-handed and right-handed β -sheets exhibit the same molecular alignment. We have also added X-ray diffraction of KLVFFAL and KLVFFAV (Fig. S6) showing that both left- and right-handed assemblies give rise to similar d-spacings both within a β -sheet and between β -sheets (see page 4 line 12 to page 5 line 2). The molecular conformation of the peptides in the two states, however, would still need to shift slightly as the right-handed β -sheets tend to have higher ϕ and ψ torsion angles as described by Chou *et al.* (Ref. 40). We have stated this on page 5 lines 7-12.

8. Another minor error: The wavelength labels in Figure 1D are reversed.

We thank the reviewer for catching this mistake in labeling. We have updated Figure 1D with the correct labels for each peak.

REVIEWERS' COMMENTS

Reviewer #1 (Remarks to the Author):

The revised manuscript answers all the queries raised by the referees.
I support the publication of the manuscript.

Reviewer #3 (Remarks to the Author):

In this revision, the authors have proven the structural transformation from left-handed manner to right-handed manner based on nice SEM experiments. The CD and FTIR data have been discussed with more detail on the analysis of the molecular conformation and arrangements. New NMR and XRD data are helpful to support the conclusions as well.

I also looked over the responses to the Reviewer #2's comments. The authors basically made a reasonable response to the concerns. What I raised is the biosafety of the amyloid supramolecular materials for the practical medical applications. This is suggested discussion with reference of the amyloid-relevant peptide assemblies for drug delivery and particularly tumor therapy etc.

REVIEWERS' COMMENTS

Reviewer #1 (Remarks to the Author):

The revised manuscript answers all the queries raised by the referees.
I support the publication of the manuscript.

We thank the reviewer for their support of this manuscript.

Reviewer #3 (Remarks to the Author):

In this revision, the authors have proven the structural transformation from left-handed manner to right-handed manner based on nice SEM experiments. The CD and FTIR data have been discussed with more detail on the analysis of the molecular conformation and arrangements. New NMR and XRD data are helpful to support the conclusions as well.

We thank the reviewer for their positive assessment of the additional data and discussion.

I also looked over the responses to the Reviewer #2's comments. The authors basically made a reasonable response to the concerns. What I raised is the biosafety of the amyloid supramolecular materials for the practical medical applications. This is suggested discussion with reference to the amyloid-relevant peptide assemblies for drug delivery and particularly tumor therapy etc.

The discussion of biosafety is an important point. We have added discussion on page 19, lines 28-43 to address this. Briefly, other amyloid-like peptides have been previously utilized for drug delivery and showed minimal systemic toxicity *in vivo*. These previous results coupled with the ability of our peptide to be degraded upon inversion and release of the drug suggest that this material has a high potential to be safe for medical applications, and that *in vivo* studies to confirm this would be needed.